# Out-of-sequence skeletal growth causing oscillatory zoning in arc olivines

Pablo Salas [1✉], Philipp Ruprecht [2], Laura Hernández [3] & Osvaldo Rabbia [3]

Primitive olivines from the monogenetic cones Los Hornitos, Central-South Andes, preserve dendritic, skeletal, and polyhedral growth textures. Consecutive stages of textural maturation occur along compositional gradients where high Fo–Ni cores of polyhedral olivines ($Fo_{92.5}$, Ni ~3500 ppm) contrast with the composition of dendritic olivines (Fo < 91.5, Ni < 3000 ppm), indicating sequential nucleation. Here we present a new growth model for oscillatory Fo–Ni olivine zoning that contrasts with the standard interpretation of continuous, sequential core-to-rim growth. Olivine grows rapidly via concentric addition of open-structured crystal frames, leaving behind compositional boundary layers that subsequently fill-in with Fo–Ni-depleted olivine, causing reversals. Elemental diffusion modeling reveals growth of individual crystal frames and eruption at the surface occurred over 3.5–40 days. Those timescales constrain magma ascent rates of 40–500 m/h (0.011 to 0.14 m/s) from the deep crust. Compared to ocean island basalts, where dendritic and skeletal olivines have been often described, magmas erupted at arc settings, experiencing storage and degassing, may lack such textures due to fundamentally different ascent histories.

[1] Departamento de Ciencias de la Tierra, Universidad de Concepción, Concepción, Chile. [2] Department of Geological Sciences and Engineering, University of Nevada, Reno, NV, USA. [3] Instituto de Geología Económica Aplicada GEA, Universidad de Concepción, Concepción, Chile. ✉email: pabsalas@udec.cl

Crystal forensics[1] or the interpretation of crystal populations and chemical fingerprints in zoned crystals is frequently applied to magmatic systems to infer processes at depth[2–7]. For mafic compositions, olivine crystals can provide access to events that occur near the liquidus of mantle-derived melts[8,9]. Therefore, olivine can track magmatic processes that may occur near the source regions in the mantle and during transit to the surface[10]. Underlying the use of crystal zoning is the assumption that the events are recorded in a sequential pattern from the oldest in the cores to the youngest at the rims. This notion of concentrically growing olivine has been recently challenged in a series of papers that propose a more complex growth history for olivine[11–14]. The presence of distinct dendritic zoning patterns in slow diffusing elements (e.g., P, Al), as well as the rare preservation of their morphologies in some olivine crystals from ocean island basalts, argue that an early dendritic olivine growth may be followed by an episode of maturation that fills the dendritic network and potentially explains the common occurrence of olivine-hosted melt inclusions. These findings have been tested experimentally on a Hawaiian basalt composition that shows that phenocryst-size (>100 μm) olivines may form within such two-stage growth for undercooling of 25–40 °C, with the crystal experiencing textural maturation as undercooling diminishes[15].

The widespread occurrence in different tectonic settings of olivine with internal dendritic arrangements of slow diffusing elements indicates that the required degrees of undercooling for the development of rapid growth textures may be a common condition among mafic compositions, which can be reached during magma mixing events[16] or when magmas experience a high thermal contrast during rapid dike emplacement in cooler crustal rocks or near reservoir edges[10,12]. While the evidence for complex olivine growth in hotspot settings has increasingly mounted, similar observations for arc settings are lacking and arc olivines continue to be interpreted exclusively through a paradigm of concentrically grown crystals. This view is driven by the polyhedral habits that dominate arc olivines where evidence for dendritic or skeletal growth representing remnants of an earlier growth stage is limited to the occasional presence of oscillatory bands of elevated P and Al zoning (e.g., ref. [17]). Such P and Al zoning alone may not be sufficient evidence to support a model of initial dendritic growth followed by crystal maturation and other textural evidence may be required to corroborate dendritic growth. For example, olivines with P and Al zoning from Shiveluch in the Central Kamchatka Depression were recently interpreted to represent concentric growth and a dendritic or skeletal growth model was not considered[18].

Here we present results combining textural analysis with fine-scale olivine chemistry and diffusion modeling to demonstrate the presence of complex growth patterns preserved in samples from Los Hornitos, a pair of Holocene mafic monogenetic cones in the magmatic arc of the Central-South Andes of Chile. Olivine textures can be classified into different growth stages ending with maturation. Those stages are partially consistent with the model of Welsch et al.[12] (Fig. 1) and correlate with compositional zoning preserved in some of those phenocrysts, allowing reconstruction of the olivine growth history. Moreover, the preservation of internal zoning constrains the time of crystal assembly and magma passage through the crust.

Unlike other volcanoes in the Andean Southern Volcanic Zone that sit on thickened crust (>45 km, ref. [19]) where erupted magmas reveal the dominance of storage and differentiation in the crust, Los Hornitos erupted primitive magma with olivine phenocrysts in equilibrium with mantle peridotite (up to Fo$_{92.5}$; ref. [20]). Magnesian olivines in arc settings have been interpreted to represent deep crystallization near or below the Moho[3,18] on the grounds of thermal considerations as well as pyroxene-barometry. In the case of Los Hornitos, structural controls may facilitate their uninterrupted ascent to the surface with limited crustal residence despite the thick Andean crust[19,20].

Previous work[20] showed that the emplacement of each cone at Los Hornitos occurred in two distinct stages. An initial explosive stage generated a widely distributed tephra fall deposit (~1.5 km around the vent) that is composed of beds of variable amounts of lapilli and ash fragments. The subsequent later stage produced lava flows that breached the earlier formed cone (Fig. S1). While the tephra of the initial stage erupted primitive bulk compositions (~14 wt% MgO; > 400 ppm NiO; ~51 wt% SiO$_2$), the later lavas are more evolved (<7 wt% MgO; < 80 ppm NiO; ~53 wt% SiO$_2$). This study focuses on the primitive tephra deposit where magnesian olivine (Fo$_{88–92}$ with Ni ~1000 to < 4000 ppm) is the dominant phenocryst phase (~18 vol%) with co-crystallizing Cr-spinel (Cr# up to 76), and subordinate clinopyroxene (<3 vol%).

## Results

**Olivine morphologies and growth sequence.** Olivines from Los Hornitos tephra are comprised of three groups that are distinguished morphologically, by size, and by their chemical signatures, most notably their Fo–Ni variations. We identified texturally that the majority of the studied olivines (total $n = 455$) are characterized by a common polyhedral habit (Group 1 and 2). Group 1 (~27 vol %) and 2 (~54 vol %) are between ~400 and 800 μm in size and we refer to them as mature olivines throughout the text (see histogram and exemplary crystals in Supplementary Information Fig. S2a, b). They differ in their degree of polyhedral habit, where group 1 olivines are fully polyhedral and group 2 olivines are transitional between polyhedral to skeletal. Furthermore, the euhedral to subhedral group 1 olivines lack open cavities. They are often imperfectly broken along one or two combined poorly-developed cleavage planes, most commonly parallel to (010) and (001). Group 2 olivines maintain open cavities in the (021) plane and less frequently also in the (010) plane.

Group 3 olivines (~19 vol %) are smaller (~100–500 μm) with externally skeletal textures. We refer to these olivines as immature as they also retain visible dendritic features in the innermost part of the crystal. The dendritic pattern of immature olivines occurs as a planar feature in the center of the crystals, along the crystallographic plane ac and is given by the occurrence of four interconnected dendrites radiating from the crystal center (Fig. 1a). Those dendrites are symmetrical and display four apices shaped as arrow-like tips toward the external part of the emerging crystal. Growth progression in the more advanced olivines leads to full enclosure of the entire external frame along this section (Fig. 1b). This dendritic arrangement is sharply bounded by four melt pockets systematically distributed along this crystallographic plane. Normal to the c axis, the initially ac elongated dendritic arms open toward the b axis, conforming the typical section of olivine, while leaving two symmetrical cavities that envelope the central crossbar along the a axis (Fig. 1c). As maturation proceeds these cavities eventually form the (021) face, which is well known to remain open even in advanced grown crystals (e.g., ref. [12]).

According to our observations, the (110) face is always present in the immature crystals and, therefore, develops early during initial olivine growth. In contrast the (021) face and, less frequently observed the (010) face, remain as skeletal cavities in the most immature olivines. These observations indicate that the earliest proto crystal is developed as a structure where the faces are limited only to its edges conforming a crystal frame, a term we will use throughout the text for this ephemeral crystal morphology (Supplementary Information Fig. S3).

Larger skeletal olivines emerge through the progressive growth of new crystal frames to the exterior. The repetitive pattern is best

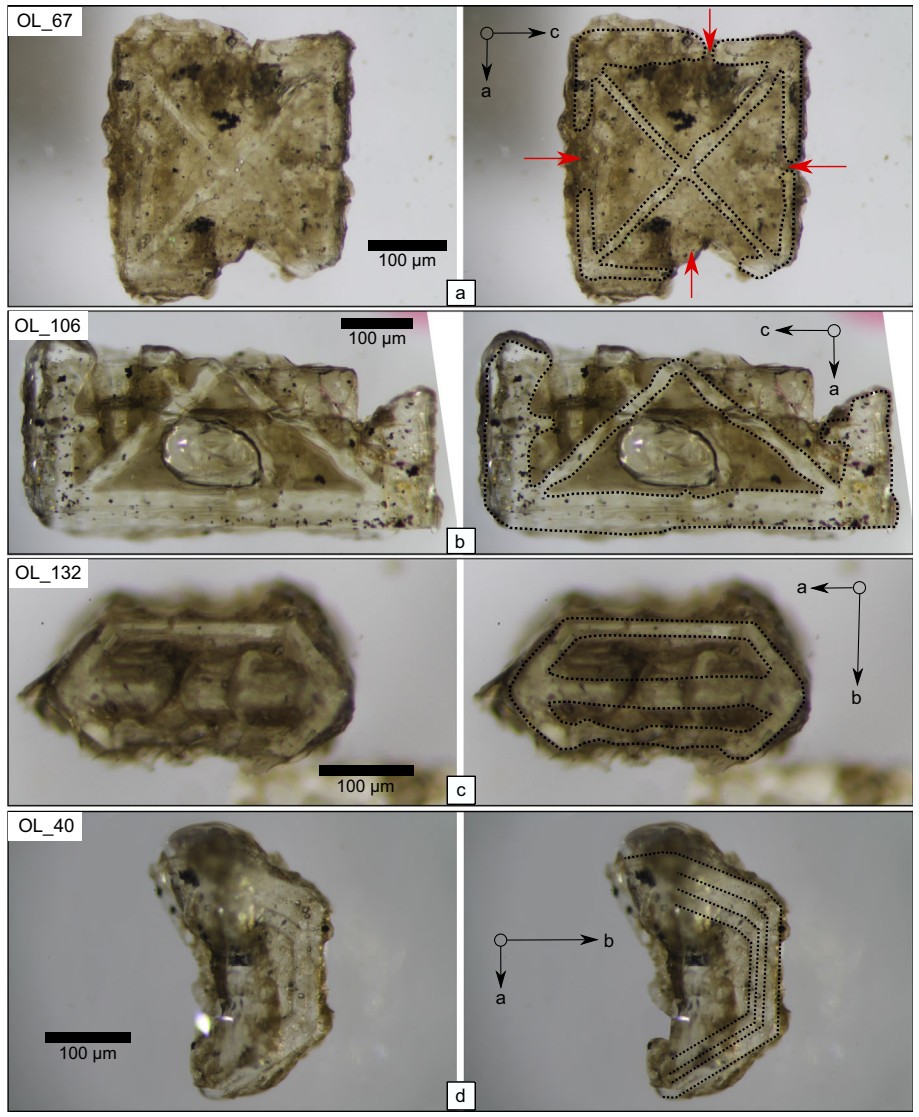

**Fig. 1 Examples of dendritic and skeletal textures preserved in olivine phenocrysts of Los Hornitos, raw images (left) and their interpretation (right).**
**a–c** Group 3 type olivines, see text for details on the groups, **a** Olivine 67 oriented normal to b axis showing the four dendrites radiating from the central core, partially enclosing the external section, leaving cavities filled with melt (red arrows) connected to the exterior of the crystal. **b** Olivine 106 broken along the (100) plane, oriented normal to b axis. This crystal is fully enclosed along the c direction and shows in detail the morphology of the dendrites and the fully entrapped melt. **c** Olivine 132 oriented normal to c axis showing the two cavities enveloping the central crossbar along the a axis. **d** Olivine 40 irregularly broken along plane (010) oriented normal to c axis. This crystal denotes the layered texture resultant of the progressive external growth of crystal frames, promoting the occurrence of a thin melt film between two adjacent growth layers. We consider this crystal part of group 2. Examples for each group are shown also in Supplementary Information Fig. S2b.

observed normal to the c axis, adding a concentric and layered texture to the initial olivine arrangement, preserving the basic structure as crystal growth proceeds. Moreover, the layered framework promotes the presence of a thin melt film between two skeletal growth layers (Fig. 1d). The small inner frames grow first and can advance toward a more mature texture, while the outer frames lag behind in a purely skeletal stage.

Compositionally, the polyhedral crystals of group 1 have the most primitive cores, reaching up to $Fo_{92.5}$ and ~3800 ppm of Ni. Group 2 olivine cores are slightly more evolved at $Fo_{91.5}$ and < ~3500 ppm of Ni (Figs. 2 and 3a). The immature olivines of group 3 share a broad similar composition as group 2 ($Fo_{91.4}$ and < ~3000 ppm of Ni) (Fig. S4). Chemical contrasts among the texturally defined groups hint that the morphology of the three groups represents a sequential record of growth, where the early more primitive olivines advance to greater textural maturation.

**Chemical evidence of growth by addition of crystal frames.** Selected olivine grains were sectioned to image elemental zoning and its potential relation to olivine growth via crystal frame addition. Detailed electron microprobe (EPMA) profiles on mature crystals of groups 1 and 2 show a broad normally zoned pattern interrupted by narrow reversals (Fig. 2). Those olivines express their compositional range as a concentric oscillatory pattern comprised of up to two frames exhibiting elevated Fo and Ni contents separated by zones of lower Fo and Ni contents. These oscillations occur on lengthscales of 10–50 µm and are more pronounced for Ni than for Fo. An analysis of the angles measured in backscattered electron images reveals that growth of those chemical oscillations reproduce theoretical angles of euhedral olivines (Fig. 2b–d). Elemental zoning obtained from rim-to-rim profiles along five different olivines (Fig. 2 and Supplementary Information Fig. S5) shows

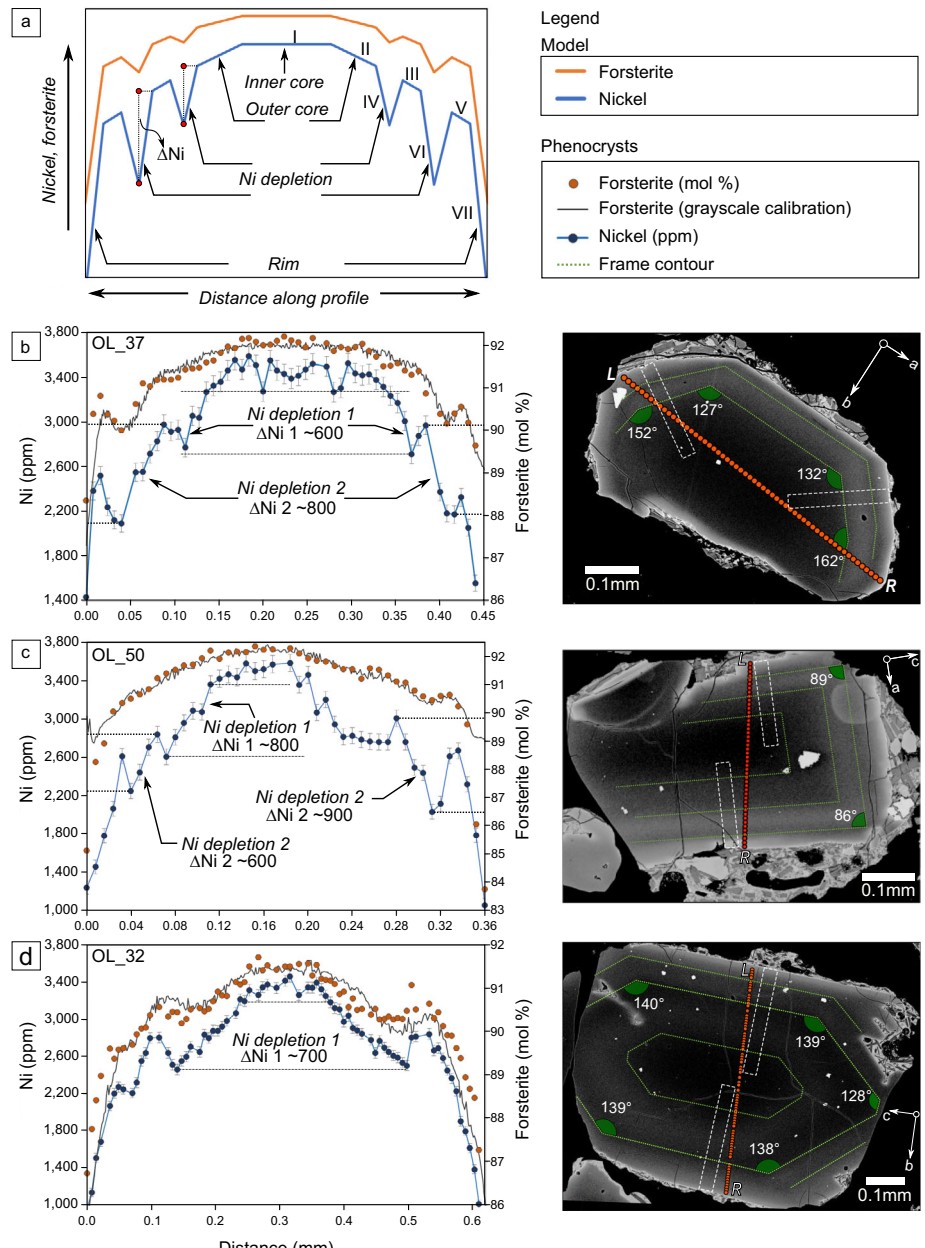

**Fig. 2 A conceptual interpretation of major (Fo content) and minor (Ni) element zoning preserved in olivine phenocrysts (group 1 and 2) from Los Hornitos. a** Schematic model of the Fo and Ni variations explained by four main zoning structures consisting of an inner and outer core, Ni depletions and rim. Roman numerals refer to the growth sequence of the crystal. **b–d** Fo (orange; right axis) and Ni (blue; left axis) contents along rim-to-rim traverses along three mature olivines sectioned normal to c, b, and a axis, respectively, with their BSE images and the analyzed traverses. Error bars on Ni data represent 1 sigma. Similar magnitudes of Ni depletions (ΔNi) are observed among the crystals. Segmented green lines above the images enhance the position of chemical reversals, which reproduce the euhedral olivine morphology, allowing the crystal to be oriented (see raw images and orientation parameters in Supplementary Information Fig. S6). White segmented blocks show the location of diffusion profiles for timescale calculations. Olivine composition data can be found in Supplementary Data 1.

qualitative agreement and also similar compositional jumps among the crystals.

The conceptual model consists of four zoning structures (Fig. 2a): an inner core of constant composition that is surrounded by an outer core that is normally zoned (I and II). Rimward, the next zone separated from the core-to-rim sequence consist of a Ni reversal (III) that merges to the central structure by a significant Ni depletion (IV) (ΔNi > 300 ppm). Major element zoning continues to be normal in this zone. Subsequent zoning continues with Ni depletions (usually two; up to ~900 ppm of ΔNi; ΔFo < 1) along the

profile, that reverse toward but do not reach contents prior to the depletion (V and VI). Lastly, the rims consist of significant normal zoning in both Ni and Fo (VII).

These patterns are symmetrical across olivine grains with only minor deviations in compositions (Fig. 2). Two exemplary mature olivines of group 1 (B: OL_37; C: OL_50) and one of group 2 (D: OL_32) have relatively flat inner cores that range between 3350 ppm of Ni (OL_32) and 3550 ppm of Ni (OL_50) with Fo contents between 91.5 and 92.3, respectively. The normal zoning of the outer cores of OL_37 and OL_50 is of similar magnitude

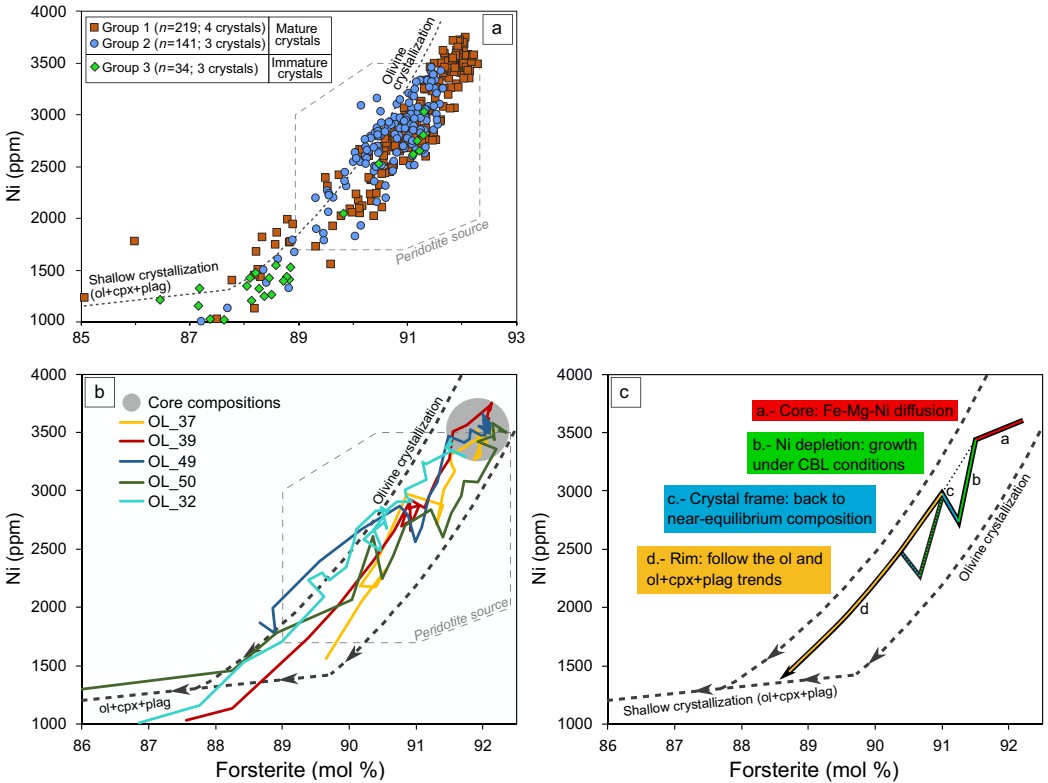

**Fig. 3 Distribution of Fo–Ni contents among the different crystal groups and variations along selected crystal profiles. a** Olivine groups separated by color show that the texturally mature crystals tend to be more primitive whereas the immature olivines are compositionally more evolved. **b** Core to rim profiles of selected crystals. Note the core compositions falling in a restricted Fo–Ni space (gray circle). Rimward, the color-coded crystals share a common evolution pattern, characterized by a sharp Ni decrease accompanied by only a slight variation of Fo, followed by abrupt increments of Ni-only variations at approximately constant Fo. Close to the margins, the crystals chemistry is consistent with co-crystallization of olivine + clinopyroxene ± plagioclase along the crystallization path. **c** Schematic illustration of the evolution pattern of a -core to rim- section of a model crystal. Labels indicate the crystallization conditions of the different slopes of segments (a–d). The alternative path in segment d shows that some crystals may experience multiple sequences of externally developed crystal frames and Ni depletions followed by a return to equilibrium crystallization, e.g., Olivine 50. CBL: compositional boundary layer. Crystallization lines after Ruprecht and Plank[3] and mantle peridotite field after Straub et al.[71].

($\Delta$Fo ~0.3 and $\Delta$Ni ~140 ppm). For these crystals, the variations of Ni associated with the inner depletion ($\Delta$Ni-1 ~700 ppm) is smaller than those of the outer depletion ($\Delta$Ni-2 ~ up to 900 ppm). OL_32 shows a more pronounced outer core ($\Delta$Fo ~1 and $\Delta$Ni ~500 ppm) and a change in Ni associated with the depletion of $\Delta$Ni-1 ~700 ppm.

Compositional variations of Fo and Ni for these olivines show a broadly common evolutionary pattern that outline the described zoning structures (Fig. 3b). The inner core Fo–Ni compositions cluster at high contents and grade into a trend with the outer core characterized by discrete variations of Ni compared to a larger variation in Fo. This trend intersects the generalized olivine crystallization path (Fig. 3c, segment a) and the shift to lower Fo contents is due to the more advanced diffusion of Fe–Mg–Ni in the cores, compared to the outer zones of the crystals (e.g., ref. [18]). In addition, the Ni depletions represent a drastic decrease of Ni that is inconsistent with olivine crystallization under equilibrium conditions (Fig. 3c segment b). Instead, this segment records the secondary effects of an episode of rapid growth where Ni is quickly depleted in the surrounding melt given the high partitioning in olivine ($Kd_{Ni}$ ol/melt > ~10 in basaltic compositions[21–25]) and the subsequent development of a compositional boundary layer (CBL) around the crystal, which is depleted in compatible elements and enriched in incompatible elements (e.g., refs. [16,26,27]). Comparatively, Fo is less sensitive to variations in the CBL environment given the higher initial concentrations of Mg and Fe in the melt and

its comparatively lower partition coefficients[23] (Supplementary Information Fig. S7). Ni reversals deviate from a broadly common value of ~2700 ppm (Fig. 3b). This segment represents the growth of a crystal frame out of the CBL environment, where near-equilibrium compositions can be resumed since the melt has not been affected by enrichment/depletion of elements out of equilibrium. By this reason, increments are more developed for Ni than for Fo (Fig. 3c, segment c). Slight positive variations in Fo, as for OL_37, indicate that the new attained equilibrium out of the CBL is comparatively less fractionated in MgO than previously in the CBL. However, these variations are < 0.6 Fo unit, whereas variations in Ni are >200 ppm. Rimward, another Ni depletion may occur, as for the case of OL_37 and OL_50, followed by a frame that detaches from a common background value of ~2200 ppm of Ni (Fig. 3b). Finally, the rims follow a trend that is more consistent with olivine crystallization, and transit toward the shallow crystallization trend (OL_50, OL_39; Fig. 3b). Fo–Ni variations are more complex in OL_32 and the core of this crystal is comparatively more evolved than the other crystals, however, in agreement with the other crystals, two major perturbations occur at ~2500 and ~2200 ppm of Ni, sharing a common growth history with the less evolved crystals.

The standard interpretation of the olivine zoning at Los Hornitos would commonly invoke magma mixing, however, we propose the out-of-sequence growth model as an alternative interpretation for the following reasons. Compositional zoning is

uniform among the olivine grains with a uni-directional evolution to more evolved compositions at the rim interrupted by short wavelength Ni depletions. Fluid motion during mixing in magmatic systems is mostly chaotic, especially in reservoirs, but potentially also in conduits. Thus, crystals nucleating and growing in a mixing, rapidly evolving environment display a great variety of compositional histories (e.g., ref. [28,29]). Some crystals may originate in the more evolved melt whereas others in the original primitive melt, leading to complex crystal populations. Moreover, during mixing, crystals may reside for different time scales in the compositionally distinct melts leading to variable lengthscales of zoning. Both of these characteristics are inconsistent with the uniform and uni-directional evolution we observe. If mixing is not only chaotic, but approaches turbulent conditions, local compositional gradients would be quickly erased producing very simple zoned crystals.

By comparing the Los Hornitos olivine zoning to other strongly-zoned olivines that are interpreted to represent magma mixing, we find important differences. The magnitudes of Fo–Ni increments associated with the growth of crystal frames ($\Delta$Fo <0.6 and $\Delta$Ni > 200 ppm) contrast with Fo–Ni variations reported for a ring tuff of Shiveluch volcano in Kamchatka[18], Llaima volcano[17] in the Andean Southern Volcanic Zone of Chile and Paricutín in Central America[30]. At Shiveluch, primitive compositions of olivine present increments of up to ~300 ppm of Ni accompanied by increments of Fo greater than two forsterite units ($\Delta$Fo > 2), whereas at Llaima volcano, evolved olivine crystals (~$Fo_{74}$) present increments of Ni on the order of ~150 ppm accompanied by increments of Fo greater than 3 Fo units ($\Delta$Fo > 3). On the other hand, chemical variations induced by the entrainment of primitive melts of distinct composition accounted for Ni reversals > ~800 ppm for mostly invariant Fo, as the case of Irazú volcano in Costa Rica[3]. A recent study on Paricutín[30] presented oscillatory zoning in intermediate Fo olivines (~$Fo_{78-86}$) and attributed those changes to complex magma mixing within the conduit during ascent. Compared to olivines from Los Hornitos those from Paricutín show much larger variations in major elemental zoning ($\Delta$Fo > 2).

And lastly, we focus in our study on the textural evolution of olivines with a significant population showing large cavities questioning a continuous core-to-rim sequence. Thus, olivines from Los Hornitos allow us to present an alternative interpretation for oscillatory zoned olivines that does not require magma mixing for the development of zoning in major, minor, and trace components.

All these observations indicate that: (1) The early structure of these olivines represent rapid growth of crystal frames disrupting a core-to-rim sequence. Those frames, composed by high Fo–Ni, merge each other by zones of lower Fo–Ni composition. Diffusion of Fe–Mg may erase such compositional reversals on timescale of weeks to months, and the return to a smooth; shoulder-less normal zoning may take years at temperatures above 1100 °C (ref. [31]). (2) The cores of mature olivines of group 1 share a similar source and maintain a unique trend of diffusion, absent in olivines of group 2 and 3, that was acquired early during its development. (3) Zoning structures within the frames are minimally affected by diffusion compared to the core of the crystals of group 1, and, therefore, they represent a recent stage of growth.

**Clinopyroxene-liquid thermobarometry.** Clinopyroxene (cpx) and spinel are the other two early crystallizing phases. Cpx occurs exclusively as microphenocryst (<300 μm) and grades into smaller sizes in the groundmass with Mg# (molar Mg/[Mg+Fe]) ranging from 81 to 86 (Supplementary Data 2). Sector zoning is commonly observed in cpx, consistent with rapid growth and development of dendritic and skeletal textures induced by a pronounced undercooling[32]. Spinels that co-crystallized in equilibrium with cpx are systematically lower in Mg# (<42) compared to the spinels that equilibrated with magnesian olivines (45 < Mg# < 76) (Supplementary Information Fig. S8 and Supplementary Data 3).

We applied the cpx-liquid barometer ($n = 24$) using the model of Al partitioning between cpx and melt (eq. 32c of Putirka[33]) and the jadeite-diopside/hedenbergite (Jd-Di/Hd) exchange model (eq. 31 of Putirka[33]). The models yield a pressure range of 4–7 kbar (14–24 km) and 3–5 kbar (12–18 km), respectively. Crystallization temperatures of 1140°–1167 °C are calculated for the cpx-melt equilibrium using eq. 33 of Putirka[33] (Supplementary Data 4). Clinopyroxene crystallized from a more evolved melt under potentially lower pressure conditions when compared to magnesian olivine because it is in equilibrium with lower Mg# spinels. Thus, the pressure and temperature estimates derived from cpx provide minimum depth and minimum temperature constraints for the assembling conditions of primitive olivines.

**Timescales.** The detailed textural investigation of the Los Hornitos olivines suggests a complex growth history where zones are not simply recording time along a continuous core-to-rim traverse. Conventionally the reversals would be interpreted to represent mixing events[2,18]. In the case of Los Hornitos, we disregard that option as olivines show multiple reversals within single crystals requiring a complex magma assembly history for this primitive magma. Furthermore, unlike other systems that show mixing of diverse mantle melts (e.g., ref. [3]), these olivines stay on a restricted differentiation path with respect to Fo–Ni systematics. Therefore, we interpret the zoning patterns representing the out-of-sequence assembly, where the crystals grow quickly in an evolving liquid from $Fo_{92}$ in the core to $Fo_{88}$ rimward (excluding the outermost rims that evolve beyond $Fo_{88}$). The growth history can then be extracted by linking up zones of decreasing Fo. As the outer frames have grown after any inner frame, they preserve more pronounced reversals. We focus our diffusion timescale calculations on these last reversals as they provide time information about two processes: (1) time between the most recent skeletal growth and eruption, where growth occurred deep in the crust (>24 km depth) and (2) given that the olivines track differentiation that has yet to be erased within the crystal zoning, the timescales also provide a unique constraint on the timing of melt differentiation in this primitive system.

Given our growth model for these olivines it is inappropriate to develop a sequential growth and diffusion model (e.g., ref. [2,14,34]), the exact initial conditions are unknown. However, the systematic stepwise lowering of each individual high Fo frame from core to rim, suggests that initial Fo contents of these zones were not significantly higher than what is preserved in individual frames. It should be noted though that while growth is still faster in these olivines than diffusion[14], particularly during rapid formation of crystal frames, growth and diffusion operate on similar timescale as crystallization leads to the merging of frames and conditions approach a Peclet number of 1 (ref. [35].). This is substantially different than olivine examples presented for Kilauea[14] where growth is instantaneous relative to diffusion.

We limit our timescale calculations to the short lengthscale reversals in three crystals for which we have good control on the orientation of the crystal section (those of Fig. 2b–d). All three crystals show reversals that are < 20 μm. The calculated diffusion timescales for these crystals range from 3.5 to 40 days (Supplementary Information Fig. S9, see also the "Methods" section for details) when we use a conservative temperature of 1150 °C for these reversals at magnesian compositions. This timescale estimate ignores effects of growth-related zoning which makes our calculations a

maximum estimate for these crystals. Previously reported timescales between formation of the zoning and eruption using Fe–Mg interdiffusion on zoned, primitive olivines from Los Hornitos are slightly longer (13–102 days)[36]. Whether this small difference is a result of the model approach or due to the fact that Tassara et al.[36] used olivines that lacked Fe–Mg reversals, cannot be fully resolved because the details on the diffusion modeling were not reported in sufficient detail. Potential differences may arise from the assumed temperature (1132 °C was used for one crystal in Tassara et al.[36]) and the differences in profile orientation relative to the crystal lattice. The previous study limited its analysis on fully matured crystals, which is deduced from their Figure 9. We specifically target crystals with measurable major element reversals, which suggests that these represent the latest olivine growth. Slightly longer timescales would be consistent with that, but it would suggest that onset of olivine growth at depth was somewhat protracted. A protracted growth history is consistent with partial diffusive equilibration of interior reversals of Fo zoning that is now represented by Ni depletions and nearly flat shoulders in Fo content (Fig. 2). Alternatively, Tassara et al.[36] reported lower diffusion coefficients that were used in the calculations compared to our calculations, which may indicate systematic differences in our approaches. Using our calculated timescale, we estimate minimum magma ascent rates on the order of 25–300 m/h (~0.007–0.08 m/s; assuming a minimum depth of 24 km constrained by cpx barometry), although those can be considerably higher if the locus of olivine assembly is deeper. Using 40 km as a conservative depth provide magma ascent rates of 40–500 m/h (~0.011–0.14 m/s; see "Discussions" below).

**Dendritic/skeletal morphologies and frozen Fo–Ni zoning testifying rapid growth regimes: an update to the olivine growth model**. Linking textural and compositional data presented above allow us to determine that Ni reversals represent zones of rapid (diffusion-controlled) growth. Locally at the frame interface the rapid growth is followed immediately by transient slower growth zones represented by the Ni depletions, due to a decrease in the supersaturation that results from the partitioning of compatible elements into the growth zones (Ni, Mg) and the concurrent increase of incompatible elements. These chemical changes may induce a slow-down of crystal growth in the boundary layer environment (e.g., ref. [12,26,37,38]). While local boundary layers may slow down growth, dendritic fast growth of other new frames may continue simultaneously, documented in the advance of multiple frames (e.g. OL_32 and OL_50).

Intensity maps for Mg, Ni, and P of two selected crystals of group 3 (OL_38) and group 2 (OL_35) (Fig. S10), show the presence of P-rich lines correlating with the broader Ni depletions. This suggests the prevalence of a transient CBL creating the depletion of Ni and the enrichment in P. The difference in lengthscale and magnitude of the narrow, higher intensity P lines, compared to the broader Ni depletions, is a result of the different diffusivities of P and Ni and that P has to be incorporated during rapid-growth, as observed by Shea et al.[16] in experimental work.

In a broader sense, we interpret the development of olivine-euhedral morphologies of elevated Fo–Ni contents that result in multiple reversals as seen in backscattered-electron (BSE) images (Fig. S6a–c) as spatial out-of-sequence growth where crystal frames form just beyond the CBL environment in order to reestablish near-equilibrium crystallization conditions. Thus, frames represent events of rapid growth that is quicker than growth rates under the CBL environment and by addition to equilibrium conditions. The result is an external frame emplaced prior to the full closure of the gap between the two successive crystal structures (Fig. S4). The out of sequence growth leaves behind a compositional suture that is represented by the minimum of the Ni depletions (Figs. 2 and 3).

These compositional gradients may be quickly obliterated by diffusion due to their short lengthscales[31]. Thus, only the outer-most, younger frames may be preserved in pronounced profiles. Similar growth features in the crystal interior get erased by diffusion. In particular, the smaller differences in Fo content are only visible in the Los Hornitos olivines as flat shoulders, while higher amplitude Ni reversals may still be preserved.

Using the findings in these natural crystals we propose an updated olivine growth model that is consistent with our textural observations and chemical zoning features. This model complements existing interpretation for zoning and textural parameters retrieved from natural and experimental samples observed elsewhere[11–15,39]. This type of olivine growth needs to be considered when interpreting olivine zoning together with the conventional interpretation of sequential core-to-rim growth; the latter being most effective under conditions of small undercooling[15].

In the out-of-sequence model, the formation of a polyhedral normal zoned crystal occurs through five main stages (Fig. 4):

Stage 1: Four dendritic extensions radiate from a crystal nucleus (Fig. 4a). During this stage, growth occurs mostly as a planar feature along the crystallographic plane ac. Those dendritic extensions enclose the entire perimeter of the crystal section leaving four pockets of melt symmetrically distributed (Fig. 4b). The particular morphology along this section is the first evidence that growth develops along preferential directions (anisotropic growth) at the expense of the whole plane of a given face, consistent with the so called Berg effect[40]. Experimental growth rate constraints[15,39,41] suggest that dendritic extensions like those described here grow on the order of minutes (~1.5 min for a ~100 microns dendrite). The crystal shape that emerges during this stage is similar to that previously described for high cooling rate experiments (1890 °C/h; $-\Delta T = 118$ °C)[42] (see the photomicrography B2 of Fig. 11 in Faure et al.[42]).

Two-dimensional (tabular) morphologies (e.g., ref. [43]), as well as preferential initial growth along the plane ac, has been also observed in experiments[15]. However, a distinctive elongation along the a axis described in these experiments was not observed in our natural samples.

Stage 2: A 3D structure begins to form when crystal growth expands along the crystallographic axis b. This growth episode initiates simultaneously from the four crystal corners, along the convergent edges of face (021) and split to conform the edges of face (010) (Fig. 4c). The advanced morphology of this stage constitutes a crystal frame and the emergence of a euhedral proto-crystal with hollow faces (Fig. 4d). As the size of the emerging crystal is limited during the initial stages (<100 μm) and by applying experimentally determined growth rates associated with the development of skeletal textures[15], we estimate that this morphology emerges on timescales of minutes to hours. The immature olivines of group 3 described above represent a textural maturation analogous to morphologies that develop during stage 1 and 2. Crystals of similar morphology have been experimentally obtained by Faure and Tissandier[44].

If saturation decreases in the silicate melt, the growth mechanism may shift to slow-growth and filling-in of cavities proceeds, thus favoring textural maturation (Fig. 4e). This is the case of the cores of mature crystals of group 1, that additionally experienced Fo–Ni diffusion.

Stage 3: Growth continues along the four dendritic extensions along ac plane (Fig. 4f) and successively, the growth of a new frame, rooted in the four recently grown corners (Fig. 4g). As consequence of this rapid growth stage, a CBL is left behind between the recently formed frame and the internal arrange-ment. Here, growth rates are comparatively slower and the melt strongly decreased in Ni (e.g., Supplementary Information

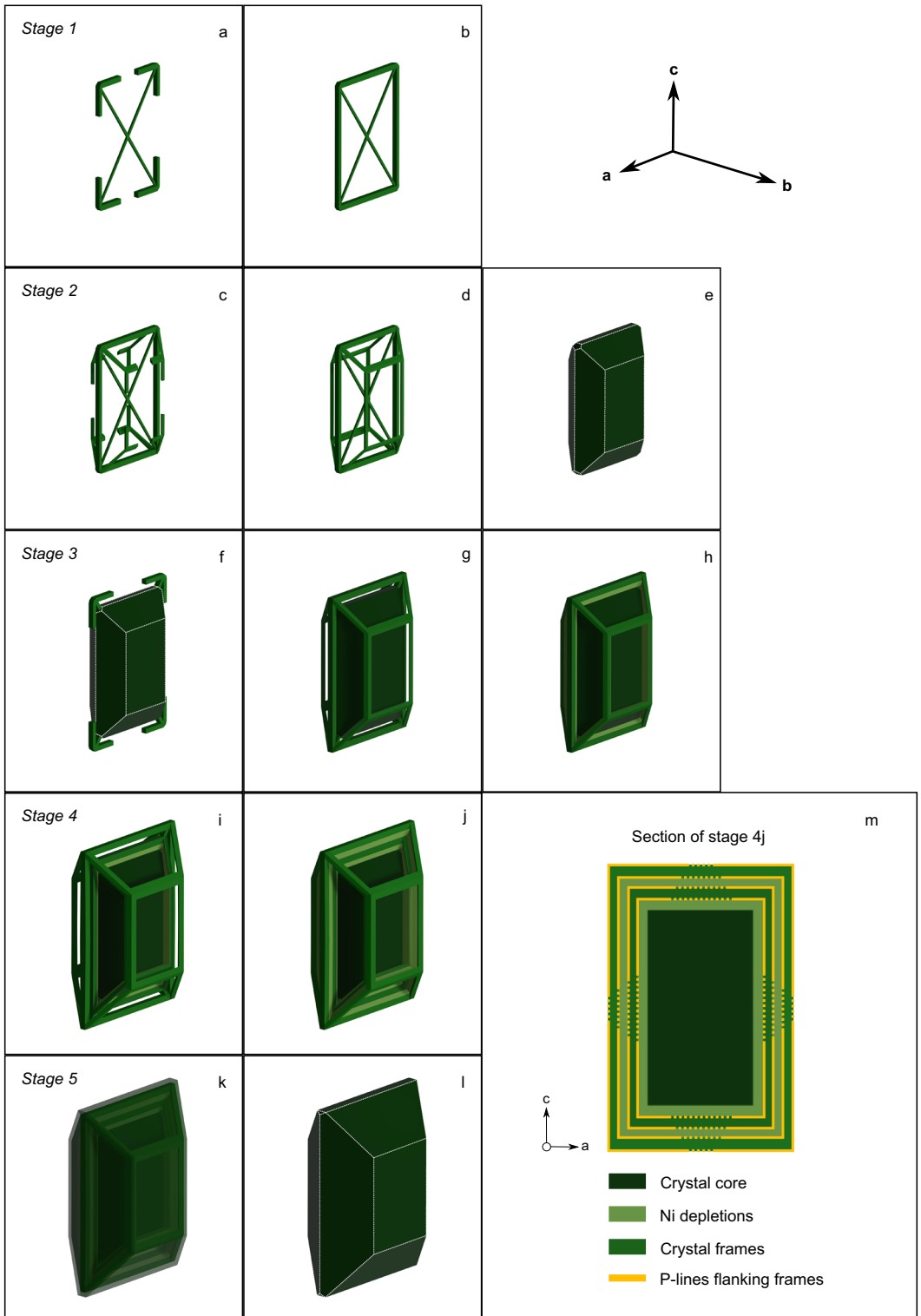

**Fig. 4 Three-dimensional growth model for Los Hornitos olivine separated into five stages. a**, **b** Four dendrites simultaneously radiate from a central nucleus along the ac plane and enclose the external section. **c**–**e** Growth detaches along the b axis simultaneously from the four corners, conforming a crystal frame, where the olivine morphology is restricted to the edges of the crystal faces. Subsequently, the recently formed inner-frame is filled as maturation proceeds and a small olivine core is conformed. **f**–**h** Crystal growth continues at rapid rates and the initial dendrites grow again along the ac plane. A new crystal frame grows separated from the internal structure leaving behind a compositional boundary layer, which is responsible for the so-called Ni depletion structure in the studied olivines (pale green structure between two successive frames in (**h**). **i**, **j** The process repeats and a new frame and associated Ni depletion is externally developed in the emerging crystal. **k**, **l** Maturation proceeds from the most internal to the outer frame and the skeletal texture transit toward a polyhedral morphology. Zoning structures may be partially or fully erased depending of the timescales and storage conditions of the host magma. **m** Crystal section along plane ac showing the distribution of P-enriched lines with respect to the location of crystal frames. Segmented versus continuous lines indicate the higher P-intensities often observed near the crystal corners.

Fig. S7), resulting in a Ni depletion segment (pale green color in Fig. 4h). Based on measurements on the rim-to-rim profiles of mature olivines of group 1 (Supplementary Data 1), the mean distance between two successive frames is ~35 μm.

Stage 4: As undercooling continues, e.g., during magma ascent into progressively cooler crust, new crystal frames can grow surrounding the previous crystal structure (Fig. 4i) with the development of contemporary Ni depletions in response to the rapidly grown frames (Fig. 4j). Some of the zones of Ni depletions are not totally filled and that explains the occurrence of elongated cavities that parallel the external crystal faces and ultimately may conform elongated melt inclusions; as frequently observed in natural and experimental crystals of olivine (e.g., ref. [12,14,15,44]).

Stage 5: Continued textural maturation leaves the innermost structures texturally more evolved. The resultant morphology corresponds to a crystal in the skeletal to polyhedral transition, as those mature olivines of group 1 and 2 (Fig. 4k). The culmination of this stage consists of a polyhedral crystal avoiding cavities, and depending on the elapsed time in the hot magmatic environment, the olivine may (or may not) record the differential composition of frames and Ni depletions (Fig. 4l).

## Discussion

Integrating textural data with fine-scale olivine chemistry allows us to illuminate the history of crystallization, from the earliest to the advanced stages of growth and maturation. At Los Hornitos those stages of growth and maturation are preserved, in contrast to other arc volcanoes where they are commonly not observed. Our model defines a concentric growth of discontinuous crystal frames, out of the core-to-rim sequence and contemporary, but slower, infill of Ni depletion zones. Our calculated diffusion timescales are associated with the later stages of crystal maturation (~Stage 4–5). These timescales cannot constrain the individual episodes of rapid growth, nor do they provide information about the integrated time of the entire crystal growth. Since internal reversals are only documented in Ni reversals, while Fo zoning is preserved as flat shoulders, growth and residence may have been protracted for weeks to months allowing partial diffusive equilibration (ref. [31]). Alternatively, interior zones experienced less efficient CBLs, potentially due to slower overall growth in response to smaller degrees of undercooling[15].

While frames represent structures of rapid growth, the depletions are a response to secondary effects of such rapid growth, where out of equilibrium conditions cause concentric zones remarkably depleted in Ni and Fo in a lesser degree. Therefore, the extent of Ni depletions in the rim-to-rim profiles (Fig. 2 and Fig. S5) is directly linked to the thickness of the involved CBL's. Our observations in mature olivines of group 1 (Supplementary Data 1), indicate that the thickness of these boundary layers is between 20 and 50 μm. Using a medium value of 35 μm and applying it to the model of Watson and Müller[45], allows us to estimate olivine growth rates on the order of $5 \times 10^{-8}$ to $7 \times 10^{-8}$ m s$^{-1}$ (Supplementary Information Fig. S7). Those values are about one order of magnitude higher than the growth rates estimated by Jambon et al.[41] and Donaldson[46] but are consistent with the 3D estimates of Mourey and Shea[15] in experiments carried out with an undercooling of 40 °C.

Since cpx barometry represent the evolution of olivine at Fo < ~88 and Ni < ~1500 ppm, the depth at which most of the olivine crystallization occurs, from Fo$_{92.5}$ to Fo$_{88}$, is deeper and hotter than the cpx constraints. Yet, the polyhedral and high Fo–Ni olivines of group 1 maintain in the core remarkable diffusion profiles (Fig. 3b, c), that are absent in the cores of olivines of groups 2 and 3. Thus, we suggest that based on compositional and thermal considerations (e.g., ref. [18]), group 1 olivines nucleated with small undercooling and later experienced diffusion in the

nominally hotter subarc mantle, whereas olivines of group 2 and 3 nucleated in the deep to mid crust (Fig. 5).

In addition, given that the olivines of group 3 share a similar textural maturation, preserving visible dendritic features along a considerable Fo spectra of 91–89, we argue that rapid growth associated to magma ascent and cooling occurred early in the deep crust and continued at mid crustal levels (~20 km depth) evidenced by the rapid-growth textures preserved also in clinopyroxene in the form of sector zoning.

These observations indicate that the host magma experienced a continuous and increasing degree of undercooling as it advanced through the cooler crust, enhancing conditions for nucleation at the expense of those more appropriate for maturation of the crystals. In this scenario, the skeletal assembly of frames around the cores occurred in the lower crust as magma moved toward the surface with nucleation and subtle textural maturation of olivines of group 3 also operating closely in time and depth (Fig. 5). Therefore, the outer frame at Fo~90.5 and ~2400 ppm of Ni, where we applied diffusion calculations (yielding timescales of 3.5–40 days), have grown and ascended from a broad depth estimate of 40 km implying magma ascent rates on the order of 40–500 m/h (0.011 to 0.14 m/s). Also, the retrieved ascent rates are more conservative than those reported by Gordeychik et al.[18] for Shiveluch (80–1200 m/h; 0.022–0.33 m/s), and is orders of magnitude higher than those reported for andesitic stratovolcanoes (e.g., 2–3 m/h; 0.0005 to 0.0008 m/s) (ref. [3]) that underwent also cumulate mobilization on ascent[34].

Our findings suggest that the presence of reversed zoning in olivine may not always reflect magma recharge and mixing. Instead, complex growth histories that lead to out-of-sequence growth can lead to reversed zoning in a system that does not experience substantial interaction with another less evolved magma. In the case of Los Hornitos, such reversals are better explained by closed-system processes such as cooling and fractionation during ascent combined with out-of-sequence growth and boundary layer effects. The result is a simplified magma assembly history for Los Hornitos and potentially other systems that show similar systematics. Small, individual batches of primitive arc basalt can reach mid to high crustal levels over short timescales. The presence of multiple magma batches stored in the crust to account for several compositional reversals recorded in olivine phenocrysts are not necessarily required. However, with the exception of small monogenetic systems, these individual batches of primitive magma rarely erupt since they often intersect crustal magma reservoirs of prolonged residence, as those underneath stratovolcanoes or in the mid to lower crust[47]. There, the early morphology and the complex zoning of olivine is erased by overgrowth and diffusion, respectively, resulting in an ascent and storage-integrated slower rate of magma transit, as in the cases of long-lived arc volcanoes[3,5,48]. Moreover, Los Hornitos is part of the Southern Volcanic Zone[49] that is the type locality for MASH processes[47]. The occurrence of monogenetic cones like Los Hornitos suggests that lower crustal MASH zones are not pervasive throughout the arc, and that crustal regions that lack zones of prolonged storage where magmas can ascent uninterruptedly to the shallow crust and eventually erupt at the surface can co-exist in thick arc crust.

Ocean island settings have been the prevalent locations where dendritic olivine growth has been described (e.g., refs. [12,14]). The scarcity of studies describing dendritic and skeletal olivine growth in other settings raises the question, whether this discrepancy is simply a sampling bias, i.e. studies have not looked for such olivines in sufficient detail, or is a result of fundamentally distinct magma storage and crystal growth regimes. Our understanding from experimental studies suggests that the formation and preservation of dendritic olivines requires rapid growth during times

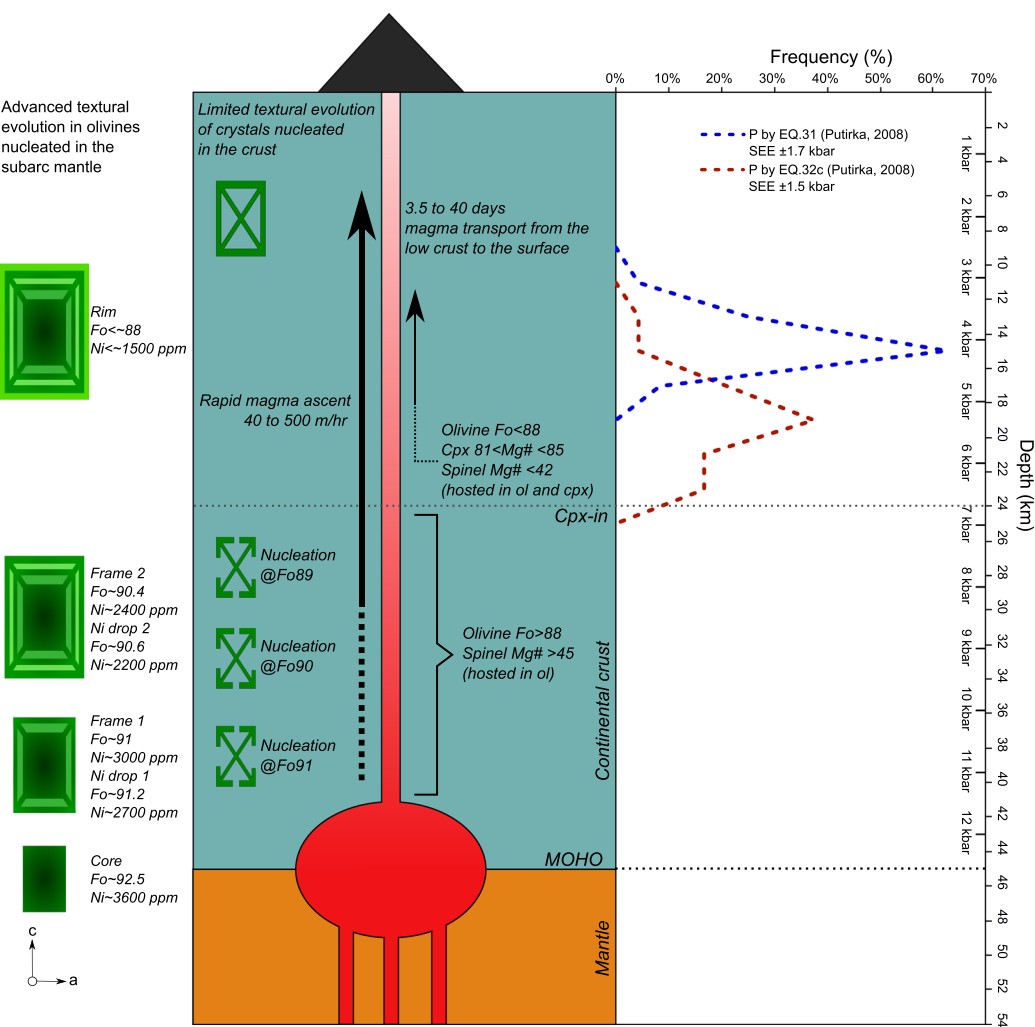

**Fig. 5 Schematic illustration of the magmatic system of Los Hornitos.** Mature olivines of Group 1, likely nucleated at mantle conditions, reach advanced textural maturity and preserve zoning structures of crystal frames and Ni depletions. Mature olivines of Group 2 and immature olivines of Group 1, nucleated in the low to mid crust, preserve skeletal and dendritic textures, and are compositionally more evolved. Diffusion calculations in the most external frame of mature olivines indicate timescales of magma transport from the deep crust to the surface on the order of 3.5–40 days. Pressure estimates are calculated using Cpx-melt thermobarometry; SEE refers to the standard error of estimate.

of large undercooling and no significant residence in the sub-volcanic magma system to limit maturation and mechanical break up of dendritic and skeletal growth structures. The latter implies rapid ascent from regions of crystal growth to the surface where textures are quenched.

Large degrees of undercooling may emerge during magma mixing, interaction of small batches of hot magma with cool crust, or during decompression-driven degassing, which leads to sudden changes in the liquidus temperature. Magma temperature changes on the order of 40–60 °C in response to mixing have been invoked[15,50]. Thus, while mixing may be the driver for crystal-lization within hotter, more mafic magmas, very large tempera-ture differences between the two mixing magmas may hinder the preservation of dendritic and skeletal olivine growth as such mixing is associated with more viscous intermediate to evolved magmas stored for extended periods in the crust. Run-up to eruption of such systems is commonly prolonged[51] likely leading to maturation of olivines and loss of dendritic features. Moreover, very large temperature gradients also cause abundant crystal-lization in the hot recharge magma and in arc settings this crystallization is dominated here by plagioclase associated with degassing and second boiling[52,53]. The large changes in viscosity

further slow-down any rapid transit to the surface and therefore the potential preservation of dendritic and skeletal olivines. Thus, magma mixing by itself may be a relatively inefficient process for dendritic olivine formation and preservation, especially in arcs. Rapid magma cooling in the crust is potentially a ubiquitous process and some studies have suggested cooling and differ-entiation during ascent[10]. However, in convergent margins with thicker crust and multiple crustal levels of magma storage[1,47] uninterrupted ascent is fundamentally hindered, which may cause the lack of preserved dendritic and skeletal olivines in arcs, though exceptions exist and they are commonly limited to small volume batches[54]. Irrespective of whether such cooling of small batches during ascent is the cause for rapid crystal growth that may favor dendritic and skeletal textures, we speculate that such effects are important but not the lone cause for the abundant occurrence of dendritic and skeletal olivine in ocean island compared to arc settings.

The fundamental difference between ocean island and arc set-tings is the volatile content dissolved prior to ascent. While ocean island basalts are commonly $CO_2$-rich and predominantly contain only small amounts of water, most arc magma volatile budgets are water-dominated. Thus, syn-eruptive degassing crystallization

pathways are distinct. Compared to water, degassing of carbon dioxide starts deep and extends over a large pressure range. Effects on the phase stability and undercooling are thus prolonged and overall much smaller for carbon dioxide degassing[55,56]. Thus, in arc settings with $H_2O$-rich but $CO_2$-poor magmas (i.e., low $CO_2/H_2O$) undercooling accelerates during degassing and abundant crystallization may hinder eruption at the surface and leads frequently to stalling and textural maturation. Only magmas with high $CO_2/H_2O$ ratios, that lead to a more continuous extraction of volatiles including $H_2O$ during crustal ascent, have the ability to experience modest undercooling and avoid substantial crystallization and stalling in the shallow crust to eventually erupt at the surface. Therefore, we speculate in the potential absence of a sampling bias that volatile content may be the first order control that prompt the preservation of textures and multi-elemental concentric chemical zoning at olivines of Los Hornitos. The continuous $CO_2$ degassing provides the driving force for a sustained magma ascent from the deep crust that is associated with the moderate undercooling and allows these discrete primitive magma batches to reach the surface. This implies a close relation between primitive mafic monogenetic volcanoes, high $CO_2$ fluxing and the preservation of dendritic and skeletal textures in olivine. The future interrogation of volatile contents in olivine hosted melt inclusions will help to answer this hypothesis and it further highlights the need to fully reconstruct volatile budgets of magmas[57].

## Methods

**Samples**. Tephra samples were collected in the fallout deposit at ~300 m from the vent. Bulk rock analysis of individual tephra layers that cover the full thickness of the fallout show broadly similar contents of major and minor elements (~12 to ~15 wt% of MgO; ~51 to ~53 wt% $SiO_2$; ~400 to ~500 ppm of NiO). We selected two representative tephra layers of fine granulometry where olivines were found as loose grains. Those samples are CLHE-TF4 (IGSN: PPRAI1021) and CLHE-TF1 (IGSN: PPRAI101Y). Samples were sieved at 25 mesh size and magnetic phases were removed. Olivine separates were obtained by the use of LST (lithium heteropolytungstate) at room temperature of 25 °C. Further, aliquots of olivine separates were mounted in epoxy as bulk grains. In addition, selected crystals were individually mounted oriented normal to the chosen crystallographic axis.

**Mineral textures**. Olivine separates embedded in mineral oil were texturally analyzed and documented under the binocular lenses. Most of the crystals are imperfectly broken along one or two combined planes, most commonly parallel to (010) and (001). Crystals preserving recognizable orientation features were hand-picked and grouped in three categories: early skeletal (i.e., preserving dendritic features in the innermost part of the crystal; Group 3), advanced skeletal (Group 2) and polyhedral (Group 1). Group 1 and 2 olivines were mounted collectively into a single mount. The similar size of these grains allowed simultaneous exposure and polishing of the cores. Each olivine from Group 3 was prepared individually, because of the delicate morphology.

**X-ray intensity maps**. Major (Mg), minor (Ni), and trace (P) element map analyses were performed in crystals OL_35 and OL_38. The intensity maps were acquired using an accelerating voltage of 20 keV and a current of 500–1000 nA on a 220 × 520 and 300 × 350 pixel area, respectively, for these two crystals, with a 1 μm/pixel resolution and a dwell time of 200 μs per pixel. The resulting maps were extracted as raw intensity data and left as relative intensity data.

**Mineral and glass compositions**. Electron microprobe analyses of olivine, pyroxene, spinel, and matrix glass were performed using a JEOL JXA-8600M Superprobe with five wavelength-dispersive spectrometers at Instituto GEA, Universidad de Concepción, Chile. Analyzed samples were prepared by the conventional technique as epoxy blocks with mounted mineral grains. The polished surface of the samples was covered with a conducting carbon coating ~25 nm thick using a vacuum evaporator JEOL JEE-4X. All analyses were performed under 15 kV using Kα lines for all elements. Natural and synthetic minerals, simple oxides, and glasses were used as reference samples, and the set of these samples varied depending on the object of analysis. Those with USNM code were donated by the Smithsonian Institution, USA (ref. [58]): diopside USNM 117733 (Si), USNM 137041 anorthite (Al), fayalite USNM 85276 (Fe), microcline USNM 143966 (K), chromite USNM 117075 (Cr), MnTiO3 (Ti and Mn), forsterite (Mg), jadeite (Na), wollastonite (Ca). Specific analytical conditions applied for different minerals and glass are described below. In all cases, corrections for the matrix effect were calculated by the ZAF method.

*Rim to rim profiles in olivine*. High contrast backscattered electron images were obtained in single olivine crystals. Rim to rim spot analyses were performed in selected olivine grains using a beam current of 20 nA (for Si and Mg) and 60 nA (for Al, Fe, Mn, Ni, and Ca). On-peak counting times were 20 s (Si, Mg), 100 s (Fe), and 120 s (Al, Mn, Ni, and Ca); high and low background count times were half the peak times for Si, Mg, and Fe, and twice for Al, Mn, Ni, and Ca. Under these conditions the 1σ detection limits were estimated to be 143 ppm Ni, 38 ppm Ca, 94 ppm Mn and 25 ppm Al. Analytical error for Si and Mg is less than 1% relative, it is <1.5% for Fe, <3% for Ni, <4.5% for Ca, and <11% for Al and Mn. Replicate analyses of the olivine standard (San Carlos olivine NMNH111312-44) were collected and averaged, and compared to reported values to evaluate the accuracy; $SiO_2$, MgO, and FeO varied from reported values by <1.5% relative. NiO varied from reported values by <6% and MnO <9%. $Al_2O_3$ and CaO are not reported for this standard, therefore, we are unable to assess the variation in our analyses for these elements. Olivines of Group 3 and OL_35 of Group 2 were measured using 20 nA for all elements. On-peak counting times were 20 s (Si, Mg) and 60 s for the remaining elements; high and low background count times were half the peak times for all elements. Under these conditions analytical error for Ni is better than 23% for all measurements except for two point with low Ni contents.

*Clinopyroxene*. Clinopyroxene was analyzed using a beam current of 20 nA. On-peak counting times were 20 s (Si, Mg, and Ca), 40 s (Al, Mn), and 60 s (Ti, Cr, Fe, and Na); high and low background count times were half the peak times for all elements. Analytical error (1σ) for Si, Mg, Ca, Fe, and Al is less than 1.5% relative, it is <5% for Na and Ti, <8% for Cr, and <15% for Mn. Replicate analyses of the clinopyroxene standard (Kakanui Augite NMNH122142) were collected and averaged, and compared to reported values to evaluate the accuracy. $SiO_2$, $TiO_2$, $Al_2O_3$, CaO, and $Na_2O$ varied from reported values by <1.5% relative. MgO varied from reported values by <2.5% and FeO <7%. $Cr_2O_3$ is not reported for this standard, therefore, we are unable to assess the variation in our analyses for this element.

*Spinel*. Spinel was analyzed using a beam current of 20 nA. On-peak counting times were 60 s for all elements; high and low background count times were half the peak times also for all elements. Chromium, V and Ti were measured using LIF crystal to minimize Ti Kβ overlap on V Kα. Analytical error (1σ) for Al, Cr, Fe, and Mg is less than 1% relative, it is <10% for Si and Mn, <20% for Ti, V, and Ni, <40% for Ca and <50% for Zn. Replicate analyses of the spinel standard (chromite NMNH117075) were collected and averaged, and compared to reported values to evaluate the accuracy; $Al_2O_3$, $Cr_2O_3$, and MgO varied from reported values by <1% relative. FeO varied by <1.6%, and Mn <55%. Manganese oxide and $TiO_2$ accuracy was determined using Fe–Ti oxide (ilmenite NMNH 96189), being both below 5% relative. $SiO_2$, CaO, NiO, and ZnO are not reported for neither of used internal standard, therefore, we are unable to assess the variation in our analyses for these elements.

Few clinopyroxene and spinel quantitative analyses were performed using a TESCAN VEGA II LSH scanning electron microscope installed in the same institution. Data were collected with a Quantax EDS system with a Bruker Nano LN2-free XFLASH® 6|3 silicon drift detector energy dispersive spectrometer (126 eV resolution for Mn Kα) and the Esprit 1.9 software. The analytical setting was 15 kV accelerating voltage and 5 nA probe current. A pure Cu standard was used for calibrating the EDS. Analyses were done under the standardless mode using PB-ZAF matrix correction method. Results are expressed as normalized wt%.

*Matrix glass*. Major and minor element composition (Si, Ti, Al, Fe, Mn, Mg, Ca, Na, K, P, S, and Cl) of glasses were determined using an accelerating voltage of 15 kV and a defocused 10 μm beam diameter. Sodium was measured first with a beam current of 4 nA to reduce alkali migration (cf.[59],). Other elements were measured with 10 nA. Peak counting time was 10 s for Na, 60 s for S and Cl, and 30 s, for remaining elements. High and low background count times were half the peak times also for all elements. Smithsonian Institution A99 basalt glass (NMNH113498-1) was used for standardization for Mg, Ca, Ti, and Fe, rhyolitic VG568 glass for Si, Al and K and Corning Glass (NMNH 117218-3) for P[58]. Sodium was standardized on jadeite, Cl on tugtupite and Mn on MnTiO3. Sulfur was measured on the sulfate peak position and was standardized on CaSO4. Related precision (1σ) based on repeat analysis of standard MRND7001 is better than 1% for Si, Al, Mg, Ca, and Na; 1.5% for Ti, about 2.5% for Fe, <3.5% for P, <9% for K, and <15% for Mn. Accuracy is less than 1% for Si and Al, about 1.5% for Fe, <3.5% for Ti and Mg, <11% for Mn and Ca, <6 for Na, <9% for K and <36% for P. Sulfur accuracy was determined on A99 glass to be <15%. Chlorine is not reported for the analyzed standards, therefore, we are unable to assess the variation in our analyses for this element.

**Diffusion modeling approach**. The images were individually processed in ImageJ. Each gray value image was calibrated using EPMA spot analyses. In a strongly zoned crystal accurate gray value calibration is difficult due to the substantial changes and gray scale over short distances. The compositional gradients are largest at the rims where normal zoning exceeds >5 mol% Fo content over less than 20 μm and edge effects further convolute the gray scale signal. In some cases, these effects lead to less accurate calibration curves for the crystal interiors with a narrow

compositional range (<2 mol% Fo), which is the main focus of the diffusion modeling. Therefore, we limited for OL_50 the calibration to microprobe spot analyses with olivine major element compositions >$Fo_{90}$. Good linear fits are recovered for the cores so that compositional profiles at accurate absolute Fo contents are extracted in ImageJ.

The diffusion modeling was performed on zoning profiles extracted from parts of the crystal where a one-dimensional (1D) profile is appropriate (center of crystal face). A region of interest significantly wider than measurement spot sizes were defined in ImageJ to smooth zoning profiles, while not altering the overall lengthscales of the zoning. We limit our timescale calculations to three crystals for which orientations are easily determined by measuring angles within the crystal and those three crystals are cut parallel to major crystallographic axes (Fig. 2). Fe–Mg interdiffusion is faster along the c-axis[60] and the Los Hornitos crystals show that oscillatory zoning is almost removed || c, retaining only flat shoulders instead of reverse zoning, and therefore limits its use for timescale estimates. All timescales are calculated from profiles perpendicular to c.

We make a number of simplifications in our diffusion calculations. The major element reversals are small (~0.2 Fo units) justifying the choice a constant average Fo content of 90.5 when calculating Fe–Mg interdiffusion. We use a temperature range of 1150–1200 °C, an oxygen fugacity estimate from the nearby Quizapu volcano[61] (mafic magmas are at NNO + 0.5; for T = 1150 °C → $fO_2 = 3.5 \times 10^{-3}$ Pa and for T = 1200 °C → $fO_2 = 1.4 \times 10^{-2}$ Pa) and the model of Dohmen and Chakraborty[60] to determine Fe–Mg interdiffusion coefficients of $9.76 \times 10^{-17}$ m$^2$/s (1150 °C and || 001) and $2.19 \times 10^{-16}$ m$^2$/s (1200 °C and || 001). Pressure was set to $5 \times 10^5$ Pa as an average crustal pressure, but it has no significant implications for the calculations.

For simplicity and given the fact that exact boundary and growth conditions cannot be extracted as well as temperature constraints introduce the largest uncertainties, we estimate timescales using an analytical solution for step function of semi-infinite constant Fo content; a choice that is made to obtain order of magnitude estimates. Note that given the short diffusion lengthscales and the limited knowledge of exact starting conditions and boundary conditions resulted from our overall olivine growth model a more detailed model is considered here to be misleading in its accuracy. Even the uncertainties related to temperature are negligible for our final conclusions and by making numerous simplifications our overall timescale estimates are constrained to within a factor of 2.

**Glass volatile contents (SIMS)**. SIMS data were collected with a Cameca IMS 7f-Geo at the Institute of Materials Science and Engineering (IMSE) in the Washington University of St. Louis (WUSTL), Missouri. A single SIMS session was devoted to volatile abundance measurements on the glass surrounding olivine crystals. The sample holder with the olivine-glass pairs were coated with a −40 nm thick- layer of gold. The procedure was adapted from Hauri et al.[62] measuring monovalent anions of $^{12}$C, $^{16}$O, $^{1}$H, $^{19}$F, $^{30}$Si, $^{31}$P, $^{32}$S, and $^{35}$Cl. The samples were pre-sputtered for 5 min. The Cs$^+$ ion beam configured using a 5–10 nA current and 10 kV of acceleration voltage was used to create a 15 μm spot size. Calibration curves were constructed using the reference glasses ALV1833-11, ALV519-4-1, ND7001, and ND6001 characterized by Kumamoto et al.[63] and Lloyd et al.[64] H$_2$O values measured on 20 external glasses range between 0.3 and 0.7 wt% (Supplementary Data 5). Given the high background during measurements for $^{12}$C for the standard mount, likely due to contamination derived from the standard mount itself, we do not report CO$_2$ data.

**Cpx-melt thermobarometry**. We applied the thermobarometric approach of Putirka et al.[65] based on the jadeite-diopside/hedenbergite exchange between clinopyroxene and the associated melt, which account a standard error of estimate (SEE) of ±1.7 kbar for the barometer and ±33 °C for the temperature determination. A re-calibration of Putirka[33] includes H$_2$O as an input parameter allowing a more suitable application to hydrous systems. These models correspond to Eq. 31 and Eq. 33 for pressure and temperature, respectively[33].

Another barometric approach that consider the partitioning of Al between clinopyroxene and melt (Eq.32c; ref. [33].) was applied to the data. This model has a SEE of ±1.5 kbar and requires temperature and H$_2$O as input parameters, with the former being provided by Eq.33 to every mineral melt pair.

Based on SIMS measurements, we used a medium value of 0.5 wt% of H$_2$O as input for barometry calculations.

We tested the chemical equilibrium conditions for clinopyroxene and the associated melt by two methods. The Fe–Mg exchange in the selected mineral-melt pairs fall into the $K_D$(Fe–Mg) = 0.28 ± 0,08 envelope[33]. As this approach does not consider equilibria between others exchange components, we also compared the predicted versus the observed DiHd component between the melt and mineral, respectively. A close match between these components validate the required cpx-liquid equilibrium for robust pressure and temperature calculations (e.g., ref. [66]).

During the calculations, we considered a Fe$^{+3}$/ΣFe ratio equals to 0.25. This value is representative of the redox conditions measured in basaltic melt inclusions of arc environment at global scale, that range between 0.18 and 0.32 (refs. [67–69]).

In order to convert the obtained values into depth, a mean value of 2.85 g cm$^{-3}$ was used for a simplified crustal rock density (e.g., ref. [70]).

## Data availability

The authors declare that the data supporting the findings of this study are available within the paper and its Supplementary Information files.

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

## Acknowledgements

The authors are grateful to Tomás Reyes for supporting to develop the 3D illustration of the model crystal. The first author was benefited by the CONICYT grant #21161063 during partial time of this research. P.R. acknowledges support from the US National Science Foundation (EAR 1426820/1719687).

## Author contributions

P.S. and P.R. carried out the field work and sampling. L.H., P.S., and O.R. performed measurements with EPMA and SEM. P.S. performed the classification and sampling of crystals under the binocular lenses, processed the analytical data, prepared the figures, tables, and schematic illustrations. P.R. performed the diffusion calculations. P.S., P.R., and O.R. prepared the manuscript. All authors participated in the discussions of this contribution.

## Competing interests

The authors declare no competing interests.
