## [Peer Review File · Nature Communications]

REVIEWER COMMENTS

Reviewer #1 (Remarks to the Author):

Review of Salas et al., 2020: The early structure of olivine from subduction zones and its missing zoning

The authors present textural and geochemical data from olivines crystals from tephra from the Los Hornitos monogenetic volcanoes in the SVZ in Chile. They combine detailed crystallographic work with extensive observations of olivine morphologies, with chemical analyses of 10 olivine crystals. These crystals are then used as proxies for the growth of olivines at different stages of the magmas' ascent from depth. First order diffusion modelling is used to provide a maximum time constraint on ascent from the region where the most primitive olivines were growing, to the surface. The paper is well written, and the morphological data is well-presented and detailed. I have a couple of main comments (numbered) with detailed line by line minor comments listed below.

It is important to note that this work is the first of its kind in olivines from subduction zones, since the advent of new geochemical and imaging facilities, however work from the 1970's laid the groundwork for this study, where it was noted that different degrees of undercooling produce different olivine morphologies. The methodology applied is sound, with sufficient detail provided in the methods section to allow replication of the study. I have made a few minor notes where some additional details should be included in the main body of the text.

1. Core compositions: the authors state that the three morphological groups have different chemical compositions; however only 10 crystals in total (maximum of 4 crystals from each group) are analysed. It is not clear from the methods whether all 10 crystals were mounted in a single epoxy block- if this has happened and given that the authors state clear differences in size between groups 1/2 and 3 there is potential that true 'cores' of mineral grains are not being analysed. More core-rim pairs from the three groups is needed to chemically classify the differences between the morphological groups to support the authors conclusions about location and rate of growth.
2. Given that extensive assumptions have been made for the Fe-Mg interdiffusion modelling, is it really appropriate to quote timescales as 3.5-40 days, when on the order of 10's of days would be more representative? These data should also be compared with the work of Tassara et al earlier within the text.

Kind regards
Katy Chamberlain

Line-by-line comments

- l. 16- are you referring to the cores of the dendritic olivines here, or all measurements?
- l. 24- Perhaps more traditional to give ascent rates in m/s rather than m/hr
- l. 42- Not so recent, see the work on olivine morphologies you cite later, in particular Donaldson 1976's work, which highlights this > 40 years ago. Also, see Faure's work from the early 2000's.
- l.60- elevated implies just overall raised concentrations- is that really what you mean, or are you referring to oscillatory zoning in P and Al within olivines?
- l.60-63- hard to read sentence, punctuation needed.
- l. 80- suggest reword to 'occurred in two distinct stages for the two cones'. By separated I assume you mean in time, but this could be made clearer.
- l. 88- Cr-spinel inclusions?
- l.88- you give a vol% of cpx; can you do the same for olivine within the tephra?
- l.92- you treat both cones the same- can you provide some justification for this? Are both cones erupted at the same time? If not, what is the time break between them? Is there any significant difference in the proportions of each of your olivine textural groups between the two eruptions?
- l.103- Fig. 1 could benefit from the addition of one additional row with an example of group 1 or 2 olivines so readers can compare the differences between the textural groups in a single figure.
- l.128- exchange lack for fall or lag?

l.129- how do you know you are analysing the true cores of these crystals, if they differ in size? Whilst a large number of analyses have been collected, a total of only 10 crystals have been analysed. Categorising the compositions of cores, based on only a maximum of 4 crystals for each group is a big assumption. More data should be collected on core/rim pairs from other exemplars of your textural end members to support this.

l.191-OL_32 line is hard to see on your Figure 3, please improve this.

l.192- Surely ruling out mixing due to not large enough differences in Fo and/or Ni assumes that the mixing components are the same? Why could mixing not be occurring between chemically-less diverse melts?

l.204- Over what timescales would these compositional gradients be erased?

l.205- More data needed to say this, see main comment above.

l.226- Could you apply various olivine-melt thermometers (e.g. Al-in-olivine; Ca-in-olivine) to compare olivine-thermometry to cpx-thermom?

l.256- Give ascent rates in m/s for comparison with literature.

Figure S7- what do the dashed orange lines and white squares relate to? Are the Fo profiles from BSE or EPMA? Why do you see differences in rim compositions between the left and right sides of the mineral grain?

l.264- in between periods of rapid growth, or during periods of rapid growths?

l.276- where is the evidence for these removed features? Without evidence this is just an assumption.

l.297- do you not see these elongations as the crystal continues to grow, and as you said evidence of any early growth is removed? So, you cannot comment on stage 1 per-se?

Section 5- It would be good to highlight what stage you attribute your timescales to within your crystal growth regime.

l.333- again not sure you see evidence for the earliest growth stages.

l.353- olivine thermometry will help test this theory of growth in mantle vs lower crust.

l.368- how were these estimates of timescales calculated?

l.369- do you mean > 102 days?

l.370- use consistent units for ascent rates

l.383- sentence needs rewording- unclear.

l.385- does overgrowth and diffusion within olivines really result in slower rates of magma ascent? I think this sentence needs flipping around.

l.388- Hildreth reference needed for MASH processes?

Fig. 2: Why are only mature olivines shown? Can you include an example of an immature olivine for comparison?

Reviewer #2 (Remarks to the Author):

There are several things that I definitively like about this study:

This work presents evidence that concentric zoning in olivine crystals can be interpreted in a different way from what is usually implied. Standard interpretation is that Mg/Fe-reversals in olivine crystals (must) indicate magma mixing involving more or less evolved magmas. This interpretation is entrenched into the current literature although several cases from hot-spot-related basaltic magmatism have been published, where zoning in slowly-diffusing clearly indicates otherwise.

This study presents a first case for such a process where skeletal growth is followed by textural maturation from an arc setting. As the authors state: "At Los Hornitos those stages of growth and maturation are preserved, in contrast to other arc volcanoes where they are commonly not observed."

With such a new and contrasted view of the growth history, additional and potentially significant new insights may be possible when analysing and interpreting such zonation.

All analytical aspects of this study are fine, the documentation of the data, the textural interpretations, and the careful approach to diffusion modelling are convincing.

With respect to the P-estimates using Putirka's calibration, I would suggest to be more careful and include the error envelope into for the extracted depth-data in figure 5.

However, there are two serious issues that need to be addressed before this work can attain the level of becoming of interest to the readers of Nature Communications:

1. The method section mentions (and the supplement documents) analytical data for more slowly-diffusing elements (compared to Mg-Fe-Ni), namely Ca and Al. Unfortunately, it apparently was not possible to analyse Cr and P. Nevertheless, it is the hallmark of earlier publications that advocated the skeletal growth-to-textural maturation crystallisation process to document this process by zoning patterns of slowly-diffusing elements. Given that other examples from arcs have made exactly the opposite observation (concentric zonation of slowly-diffusing elements), it is a major problem in this study that such elements have not been analysed or not been used to make their case for early skeletal growth. Group 1 olivines in this study should show concentric Al-zonation and group 2 and 3 crystals should show "out-of-sequence" zonation for Al. This test has not been made here and this is really unfortunate.

2. While I am prepared to accept the interpretation for the three-stage growth history of olivine crystals in this study, I wonder if not wider implications could be made. Is it that such detailed observations have not been made in basaltic arc magmas or is it that arc magmas indeed only rarely show such olivine growth histories. In the first instance, unfortunately, not much can be said. However, published and unpublished data indicate that such skeletal olivine growth may indeed be more common than previously thought in arcs, even on super-thick crust as in the Central Andes (e.g. Mattioli et al, 2006, J Volcanol Geotherm Res. 158: 87 – 105). If indeed skeletal growth is more typical in hot spot basalts, than in arcs, this difference must be explained. Maybe it is simply the different magma production rates and water contents that govern degassing and undercooling and the predominance of basalts in hot spots over more evolved compositions on arcs where primary mantle-derived basalts are occur more rarely.

In both cases, the question must be raised what the different growth histories may indicate with respect to different degrees of supercooling and how this may relate to systematically different magma potential temperatures, superheating of magmas, different water contents and effects of degassing and melt accumulation and ascent processes. All of these parameters will be different between hot spot basalts and arc basalts and a deeper consideration of such wider implications should be able to significantly enrich the present study. The manuscript touches on this issue in several instances but never goes to any depth.

So, in essence, I think that this study has great potential and eventually could become publishable in Nature Communications, but it clearly needs the consideration of wider context.

(BTW, it also needs a more catchy title...)

Gerhard Wörner

Reviewer #3 (Remarks to the Author):

This article brings together information from both textures and diffusion patterns of olivine phenocrysts from a monogenetic centre in Southern Andes. From the different stages of maturation and compositional reversals in those crystals the authors propose an evolutive pattern that is more complex than expected and notably preserves early stages of olivine nucleation. The latter is significant because upper crust differentiation of arc basalts usually deletes the early structure as crystals interact with impending magmas or they just stall in low pressure conditions. Taken into account this complexity the authors identify diffusive zones and use them to estimate timescales and ultimately a very fast ascent through the crust. The authors make a good selection of the case-study, take a novel approach combining detailed textural analysis and compositional gradients and thus shed light on a usually elusive stage of crystal growth. The short residence time is somewhat expected in mafic magmas from minor centres but only quantitatively estimated in a few cases yet.

Some specific questions/comments are listed below:

(L78). Although beyond the scope of this article, 'structural control' itself seems to be not enough to explain mafic magmas from Los Hornitos crossing the crust without interruption. Quoted references in fact show other monogenetic centres under a similar 'structural control', but slightly more evolved. Is there any other potential reason for such a primitive signature?. Can be the other mafic magmas from monogenetic centres modelled with a common mantle source and melting conditions?

(L86). The study focuses on the tephra sequence related to the younger vent, where magnesian olivines dominate, but it would be good to know more about the sequence associated to the older cone. The latter is the first dike ascending so a comparison could inform on relative ascent velocities, which could be higher for the second if the channel was already established. A direct test in these two sequences from the same vent (but probably close in time) would be an additional demonstration of the growth model proposed (with other constraints remaining the same).

(L237). What is the actual Fo range in the core to rim sequence?. What the meaning of the outermost rims that evolve beyond Fo88?.

(L250). What is implied in physical terms from the difference between growth of crystals from Kilauea and those from Los Hornitos?. Magma ascent is thought to be fast in monogenetic centres (and thus has been estimated in this article) and hence no significant difference would be expected with basalts from oceanic islands, especially for the rejuvenated stage. Could be this Peclet number telling something about the early crystallization stage before immediately the ascent?

In this document we are providing a detailed response below to each comment of reviewers. As a result of the revisions, we have included the following additional material to the manuscript and supplementary material:

- **A discussion on the importance of skeletal growth in arc settings compared to others, primarily ocean islands (Reviewer #2 and Reviewer #3).**
- **We added new data in the form of elemental maps (Ni, P, Mg) to shed light on the relation between fast (Ni) and slow (P) diffusing elements, and how that fits with the traditional observation of slow diffusing elements as the hallmark for complex growth (Reviewer #2). We also updated our growth model to include these conclusions (Fig. 4m).**
- **The supplement was expanded with a detailed description of the location site of Los Hornitos (Fig. S1), as requested by Reviewer #1. In addition, we included Fig. S4 to highlight the delicate morphology of dendritic olivines of group 3 and show the location of EPMA spots on those (Reviewer #1).**
- **We included the reference of Brehm and Lange (2020) that illustrate supporting conclusions of olivine growth in route to the surface as we suggest for Los Hornitos.**
- **We added in particular the reference of Albert et al. (2020), which interprets similar zoning patterns as within dike mixing and differentiation. Some fundamental differences exist between their and our study, which we highlight. We are not evaluating their model in detail. It is based on significantly less textural work. The textures are a key component in our arguments to develop the out-of-sequence growth model.**
- **We also made minor editorial edits in the main text that were not included in this response, but are highlighted in the manuscript with tracked changes.**
- **In total the length and number of references of the manuscript increased due to the various requests from the reviewers.**

Response to Reviewer #1 Dr. Katy Chamberlain

Review of Salas et al., 2020: The early structure of olivine from subduction zones and its missing zoning

The authors present textural and geochemical data from olivines crystals from tephra from the Los Hornitos monogenetic volcanoes in the SVZ in Chile. They combine detailed crystallographic work with extensive observations of olivine morphologies, with chemical analyses of 10 olivine crystals. These crystals are then used as proxies for the growth of olivines at different stages of the magma's ascent from depth. First order diffusion modelling is used to provide a maximum time constraint on ascent from the region where the most primitive olivines were growing, to the surface. The paper is well written, and the morphological data is well-presented and detailed. I have a couple of main comments (numbered) with detailed line by line minor comments listed below.

It is important to note that this work is the first of its kind in olivines from subduction zones, since the advent of new geochemical and imaging facilities, however work from the 1970's laid the groundwork for this study, where it was noted that different degrees of undercooling produce different olivine morphologies. The methodology applied is sound, with sufficient detail provided in the methods section to allow replication of the study. I have made a few minor notes where some additional details should be included in the main body of the text.

1. Core compositions: the authors state that the three morphological groups have different chemical compositions; however only 10 crystals in total (maximum of 4 crystals from each group) are analysed. It is not clear from the methods whether all 10 crystals were mounted in a single epoxy block- if this has happened and given that the authors state clear differences in size between groups 1/2 and 3 there is potential that true ‘cores’ of mineral grains are not being analysed.

R: Thanks for pointing this out. We perfectly understand the concern of the reviewer regarding this point and during the time-consuming textural work we continuously questioned any grouping. However, we are confident that these groupings are meaningful and based on a combination of careful work on individual crystals that commonly break through the center, a size consideration, textural observations of 455 olivines and an additional observation of ~200 randomly oriented olivine crystals analyzed for their composition irrespective to any specific crystal orientation (not published here because we think a texturally controlled dataset is important in this manuscript).

In detail: We clarify in the new manuscript that group 3 olivines were mounted and analyzed separately because they were delicate crystals that were treated individually. Furthermore, these crystals are broken along common cleavage planes, that run through the crystal center and from where dendritic arms radiate (Fig. S4). Once mounted these crystals were only lightly polished, and thus, we are confident that centers were exposed. Their dendritic nature resulted in only small central “cores”.

The precise crystal orientation for OL_32, OL_37 and OL_50, included in Fig. 2 of the manuscript, was obtained by measuring the interfacial angles. We did not have EBSD available at the University of Concepción to determine their orientation via a different method, but the exceptional agreement with predicted angles throughout our microscopic analysis of several crystals suggests that these orientations are robust. The two of the six remaining crystals (OL_34 and OL_39) were not properly set during mounting, which led to imperfect exposures of a crystal plane. In addition, OL_34 was polished in excess and the true core already has been removed. These careful preparations, excellent agreement of angles allow us to say confidence, where we “are” within these crystals.

Furthermore, if they were off-center cuts that simply miss the core, one can use Fig. 2 to estimate how much of the crystal on average would be missing, using the maximum Fo in the interior of group 3 olivines. If all the high $>Fo_{91.5}$ core that is making up group 1 cores (often more than 60% of the crystal diameter) were accidentally lost/not exposed in the preparation, then adding this part back into crystals of group 3 would make them substantially larger than group 1 and 2 olivines.

Lastly, note that olivines from group 3 are texturally very distinct and any concern of missing true core compositions for this group is neglecting the textural differences. The crystals of this group lack a well-developed core and maintain distinct dendritic features in the innermost part that are readily distinguishable of the more mature olivines of group 1 and 2. Olivines of group 3 were analyzed along the dendritic extensions parallel plane *ac*. We included a new figure in supplementary to show these analyzed features (Fig. S4).

*** Note we have included 2 additional points analyzed in OL_106 of group 3. Such points were updated in Fig. 3a and Supplementary Table S1.**

More core-rim pairs from the three groups is needed to chemically classify the differences between the morphological groups to support the authors conclusions about location and rate of growth.

R: While we recognize that more core to rim analysis may indeed help to better discriminate the compositional contrasts among olivines of group 1 and group 2, that in fact is not

significantly changing the model presented here. It is worth to note that the texturally more evolved olivines are enclosed by well-developed flat external faces (group 1, polyhedral crystals in Fig. 2 b, c), whereas OL_32 (group 2, Fig. 2d) remains with an open cavity in the (021) plane, evidencing the lesser degree of textural maturity. Dendritic olivines are undoubtedly different since they lack a well-developed core and present unique dendritic extensions along plane *ac*.

The main requisite for such different degrees of textural maturation is time, which is something that was also recognized and argued for in a recent experimental study (Mourey and Shea, 2019). Thus, the polyhedral crystals represent the earliest nucleation event whereas the dendritic olivines represent the latest of the three nucleation events. Such textural observations are supported by chemical contrast among the core compositions of group 1 and 2, but also by the particular diffusion profile in the cores of olivines of group 1 which is absent in group 2 and 3. While only a subset of the olivines was chemically characterized, the overall work includes the textural analysis of $n=455$ olivines. This carefully collected textural information allows us to be selective in our compositional data collection. Therefore, we are confident that the existing compositional and textural data allow us to discriminate between these three different groups.

2. Given that extensive assumptions have been made for the Fe-Mg interdiffusion modelling, is it really appropriate to quote timescales as 3.5-40 days, when on the order of 10's of days would be more representative? These data should also be compared with the work of Tassara et al earlier within the text.

R: Regarding the choice of timescales estimates: We significantly extend this discussion in this new text. We continue to use the range of 3.5-40 days, because it is a conservative (i.e. maximum) estimate from our calculations. Our low T estimate (1150°C) assumes all zoning is produced by diffusion, and suggests that growth may act on similar timescales which would shorten any timescale calculation. A careful review of Tassara et al. (2020) with respect to their modeling details, left us uncertain in their details about their modeling assumptions. They only report a single crystal (their Fig. 9) with information about lattice orientation, profile orientation, and major element zoning. The text itself provides limited additional information. The only other information is provided in their Table 4. The table summarizes their timescale results. It does not provide details about the parameters going into the calculation (only the final diffusion coefficient is given) nor does it provide information about the zoning lengthscale and style of zoning. Therefore, we discuss those aspects in the new version, but refrain from making a final judgement on how these two studies compare. At large, the results are similar and potential small systematic differences between Tassara et al. and our study have no significant impact on our findings. Regarding the estimated ascent rates: these are strictly calculated from our estimates (which we can better assess), but given the simplicity in extracting ascent rates any reader quickly will be able to assess whether they agree with our discussion or not.

In summary, the comparison between Tassara et al. 2020 and our study shows how careful our community should be when taking these timescales as definitive results. Model assumptions easily affect the reported time scales, and we state this directly in the extended methods section that any such estimates are likely uncertain to a factor 2 (line 630).

We have incorporated the suggestion to cite Tassara et al. (2020) earlier in the “timescales” section.

Page 8: lines 255-280, new text:

“We limit our timescale calculations to the short lengthscale reversals in three crystals for which we have good control on the orientation of the crystal section (those of Fig. 2b, c, d). All three crystals show reversals that are <20 μm . The calculated diffusion timescales for these crystals range from 3.5-40 days (Supplementary Fig. S9, see also the Methods section for details) when we use a conservative temperature of 1150°C for these reversals at magnesian compositions. This timescale estimate ignores effects of growth-related zoning which makes our calculations a maximum estimates for these crystals. Previously reported timescales between formation of the zoning and eruption using Fe-Mg interdiffusion on zoned, primitive olivines from Los Hornitos are slightly longer (13-102 days)³⁴. Whether this small difference is a result of the model approach or due to the fact that Tassara et al.³⁴ used olivines that lacked Fe-Mg reversals, cannot be fully resolved because the details on the diffusion modeling were not reported in sufficient detail. Potential differences may arise from the assumed temperature (1132°C was used for one crystal in Tassara et al.³⁴) and the differences in profile orientation relative to the crystal lattice. Given that we specifically target crystals with measurable major element reversals, suggests that these may represent the latest olivine growth, while the previous study limited its analysis on fully matured crystals, which is deduced from their Figure 9. Slightly longer timescales would be consistent with that, but it would suggest that onset of olivine growth at depth was somewhat protracted. A protracted growth history is consistent with partial diffusive equilibration of interior reversals of Fo zoning that is now represented by Ni depletions and nearly flat shoulders in Fo content (Fig. 2). Alternatively, Tassara et al.³⁴ reported lower diffusion coefficients that were used in the calculations compared to our calculations, which may indicate systematic differences in our approaches. Using our calculated timescale, we estimate minimum magma ascent rates on the order of 25 to 300 m/h (~0.007-0.08 m/s; assuming a minimum depth of 24 km constrained by cpx barometry), although those can be considerably higher if the locus of olivine assembly is deeper. Using 40 km as a conservative depth provide magma ascent rates of 40 to 500 m/h (~0.011-0.14m/s; see discussions below)”.

Kind regards
Katy Chamberlain

Line-by-line comments

l. 16- are you referring to the cores of the dendritic olivines here, or all measurements?

R: We are not sure how to address this comment. At the beginning of line 16, in parenthesis, we refer to the broad composition of the cores of polyhedral olivines (group 1), and further down, beginning of line 17, we refer exclusively to the composition of dendritic olivines (group 3), which holds the lower forsterite and nickel contents among the three groups of crystals. Note in the new manuscript lines 15 and 16 contain that information.

l. 24- Perhaps more traditional to give ascent rates in m/s rather than m/hr

R: We have looked into the literature and find references using either of those ascent rates

units. We decided to keep m/hr through the text since in our opinion it is more intuitive to readers when considering crustal scale processes. However, if the editor or reviewers are strictly requesting the use of m/s, we are happy to change that. We have included m/s in parenthesis for any mention of ascent rates.

1. 42- Not so recent, see the work on olivine morphologies you cite later, in particular Donaldson 1976's work, which highlights this > 40 years ago. Also, see Faure's work from the early 2000's.

R: Thanks for suggesting these two references, we had previously looked at both, including a large number of contributions of Faure and collaborators regarding olivine growth.

We reckon the work of Donaldson (1976) as relevant in terms of the different olivine morphologies obtained with experimental work. However, the link to natural samples within the community has significantly changed our view with the work of Milman-Barris et al. (2008). Their use of slow diffusing elements (P, Cr, Al) incorporated in natural olivine provided new evidence for complex growth of olivine, specifically the dendritic and skeletal morphologies. Thus, we summarize the existing work from Milman-Barris et al., 2008 onwards.

Page 2: line 33.

1.60- elevated implies just overall raised concentrations- is that really what you mean, or are you referring to oscillatory zoning in P and Al within olivines?

R: Thank you for this observation. We refer here to the oscillatory zoning of P and Al within olivine. We clarified this.

Page 2: line 50.

1.60-63- hard to read sentence, punctuation needed.

R: Thanks for this suggestion, we have revised this sentence. This sentence was also highlighted by reviewer #2, and therefore we used both reviewer comments to reword this sentence.

Page 2: lines 51-55, new text:

“Such P and Al zoning alone may not be sufficient evidence to support a model of initial dendritic growth followed by crystal maturation and other textural evidence may be required to corroborate dendritic growth. For example, olivines with P and Al zoning from Shiveluch in the Central Kamchatka Depression were recently interpreted to represent concentric growth and a dendritic or skeletal growth model was explicitly rejected¹⁸”.

1. 80- suggest reword to ‘occurred in two distinct stages for the two cones’. By separated I assume you mean in time, but this could be made clearer.

R: Thanks for this observation. We have revised this paragraph and it is now clearly expressed. In addition, we have included a new figure in the Supplementary material to give more details of the setting of the cones (Fig. S1).

Page 3: lines 72-73, new text:

“Previous work²⁰ showed that the emplacement of each cone at Los Hornitos occurred in two distinct stages”.

1. 88- Cr-spinel inclusions?

R: Indeed, those are Cr-spinel inclusions. Here we refer to the fact that Cr-spinel is a co-crystallizing phase along with high-forsterite olivine. Therefore, we prefer not to include the word “inclusion”. The writing was revised and improve clarity. In addition, we have recalculated the approximate mineralogy to the bulk tephra contents instead of reporting mineral abundances from separates, which we previously reported.

Page 3: line 77, new text:

“This study focuses on the primitive tephra deposit where magnesian olivine (Fo₈₈₋₉₂ with Ni ~1000 to <4000 ppm) is the dominant phenocryst phase (~18 vol%) with co-crystallizing Cr-spinel (Cr# up to 76), and subordinate clinopyroxene (<3 vol%)”.

1.88- you give a vol% of cpx; can you do the same for olivine within the tephra?

R: We have included the olivine percent within the tephra samples. See also the previous response to our comment to line 88.

Page 3: Line 79.

1.92- you treat both cones the same- can you provide some justification for this? Are both cones erupted at the same time? If not, what is the time break between them? Is there any significant difference in the proportions of each of your olivine textural groups between the two eruptions?

R: We have completely revised this paragraph and we have added a site description by including Figure S1 in the supplementary material. Based on the preservation state and stratigraphic relations among the eruptive products, we know that the East cone is younger but the time elapsed between the two eruptions is not known (there is age information known C-14, but it has not been published yet). The tephra deposits related to the older cone are only locally preserved and bulk-rock analysis are similar to those well-preserved of the younger cone (~14 wt% MgO; ~500 ppm NiO; 51 wt% SiO₂). We anticipate a similar crystallization history in the olivines of the older cone. However, this is beyond the scope of this work.

Combining the comments to lines 80-92 the paragraph now reads:

Page 3: Lines 72-80

“Previous work²⁰ showed that the emplacement of each cone at Los Hornitos occurred in two distinct stages. An initial explosive stage generated a widely distributed tephra fall deposit (~1.5 km around the vent) that is composed of beds of variable amounts of lapilli and ash fragments. The subsequent later stage produced lava flows that breached the earlier formed cone (Fig. S1). While the tephra of the initial stage erupted primitive bulk compositions (~14 wt% MgO; >400 ppm NiO; ~51 wt% SiO₂), the later lavas are more evolved (<7 wt% MgO; <80 ppm NiO; ~53 wt% SiO₂). This study focuses on the primitive tephra deposit where magnesian olivine (Fo₈₈₋₉₂ with Ni ~1000 to <4000 ppm) is the dominant phenocryst phase (~18 vol%) with co-crystallizing Cr-spinel (Cr# up to 76), and subordinate clinopyroxene (<3 vol%)”.

1.103- Fig. 1 could benefit from the addition of one additional row with an example of group 1 or 2 olivines so readers can compare the differences between the textural groups in a single figure.

R: Thank you for this suggestion. So far we have kept the figure as is, focusing on the dendritic and skeletal patterns in the main manuscript figure. We show an olivine of group 2 in Fig. 1d for comparison. It is comprised by at least three crystal frames that have not been totally merged yet. In addition, we include a comparative figure with examples of each group

in the supplementary Fig. S2b. We have not included polyhedral group 1 crystals in the main figure to conserve space. We refer now in the figure caption of Fig. 1 to Fig. S2b to guide any reader to that comparison if they want to see more examples side by side.

1.128- exchange lack for fall or lag?

R: Thanks for this observation, we have made the change.

Page 4: Line 119.

1.129- how do you know you are analysing the true cores of these crystals, if they differ in size?

R: The size differences are small as discussed above. Of course, we recognize that the core can be intersected at different depths, either close to the geometrical center or slightly off-center. However, note that the basis for discriminating the three groups is texture. For instance, Fig. S2b in supplementary material, show in a and b, and c and d, mature olivines of group 1 and 2, respectively. While group 1 olivine are completely enclosed by polyhedral faces, group 2 shows a comparatively lesser textural maturation, maintaining a cavity in 021 plane. These textural observations are consistent with a slightly more evolved composition of the cores, suggesting that group 2 olivines nucleated in a different event. As earlier mentioned, an exclusive feature of group 1 olivine, consist in a distinct diffusion profile in the innermost core, absent in group 2 and group 3 olivines. Even if the group 3 is a little smaller, there is not much additional core that could be added. We have expanded the answer to this issue in the answer to the main comment #1.

Whilst a large number of analyses have been collected, a total of only 10 crystals have been analysed. Categorising the compositions of cores, based on only a maximum of 4 crystals for each group is a big assumption. More data should be collected on core/rim pairs from other exemplars of your textural end members to support this.

R: We understand that more data is always preferred. However, we are limited in our ability to collect more data. Given that the categories are primarily derived from textures, our chemical patterns are only supporting evidence. The main model is much more reliant on our careful textural analysis ($n=455$). Also, we would like to note that reviewer 2 and 3 has not requested this kind of additional data.

1.191-OL_32 line is hard to see on your Figure 3, please improve this.

R: Thank you for this observation, we changed the line of OL_32 in Fig. 3 to have it stand out more clearly.

1.192- Surely ruling out mixing due to not large enough differences in Fo and/or Ni assumes that the mixing components are the same? Why could mixing not be occurring between chemically-less diverse melts?

R: We agree, reversals like those observed in olivines of Los Hornitos have been traditionally interpreted to represent magma mixing. And co-author Ruprecht has published on that in various publications. Mixing is clearly a potential mechanism to generate reversals. In complex magma systems it would not be possible to rule it out. However, here multiple reversals (see Ni in Figure 2) would require an exceptionally complex mixing history for an otherwise simple system. Moreover, those individual zones in Fig. 2 are all different compositionally and would require not just two distinct melts, but several more.

In fact, a model employing mixing explaining “oscillatory” zones olivines was recently proposed for Paricutin (Albert et al. 2020). These olivines show much larger compositional variations at more intermediate Fo contents (core vary between Fo78-85, reversals are large >4 Fo units). Their model ultimately requires an exceptionally difficult ascent history. While that may be possible and maybe more likely in a system of more intermediate composition, we provide an important alternative model that highlights the need to further constrain how zoning emerges and what it means. We have included this work now in our discussion to contrast our interpretations with theirs. Overall, our work uses a much larger population of studied olivines with respect to their textures. The presence of large cavities (not reported for Paricutin) makes our model at least equally viable. We thus think that neglecting textural observations on a large population of olivines prior to the chemical characterization may lead to an erroneous interpretation.

In our opinion, there are two main evidence to rule out magma mixing:

1.- Very restricted time in the crust: the timescales obtained in the reversals indicate < 40 days of residence in the crust (likely less than 10 days). The scenario of magma mixing with such primitive melts is likely going to happen in the deep crust, restricting even more the time rates to occur magma mixing. We find all groups/textures in a single eruptive layer. If mixing (as suggested for Paricutin) were important, we would expect that a single eruptive horizon should be dominated by just oscillatory zoned olivines. The presence of all types rather suggests a sequential formation. And again, the occurrence of at least three different reversals all leading to different compositions in the crystals would require multiple mixing events with each mixing event sequentially more evolved melts to generate the observed reversals.

2.- Ni depletions: these zones in the olivines of Los Hornitos record a period of growth under the effects of a chemical boundary layer, where Ni contents are uncoupled from the equilibrium composition for the given Fo (see Fig. 3b). The steep character of the Ni depletions is evidence that these zones are not rims in contact with an external melt that was entrained by a less evolved component, but rather these Ni depletions are better explained as rapid growth-related features.

1.204- Over what timescales would these compositional gradients be erased?

R: We agree this is an important question to consider. First, it is important to ask what “erasing” actually means. Is it the removal of the reversal, the removal of any shoulder that may indicate the former existence of reversals, or the complete equilibration? We think the former two timescales are interesting here, and we have adjusted our statement in line 203. Assuming the same thermal conditions (1150 °C) and no significant differences in oxygen fugacity (pressure dependence is overall negligible for this question), we can do some forward modeling with a rim composition of <<90 (i.e. the final zoning). The gradients are largest and fluxes are highest with such rim compositions. It takes weeks to months to remove reversals (the crystal will still be left with flat shoulders – so a somewhat complex zoning remains visible) and years to return to a normally zoned olivine.

Page 6: Line 202-204

1.205- More data needed to say this, see main comment above.

R: We think that we have address the general concerns about more data above.

1.226- Could you apply various olivine-melt thermometers (e.g. Al-in-olivine; Ca-in-olivine) to compare olivine-thermometry to cpx-thermom?

R: We have applied the Al in olivine thermometer of Coogan et al. (2014) to our olivine and Cr-spinel data sets, and we obtained estimates of 1124 to 1188°C for the forsterite range Fo89.4 to Fo91.25.

However, we consider these calculations as broad determinations since there are limitations to the application of that thermometer to our samples:

1.- The calibration is made for high temperature magmas in LIP's or OIB for the temperature range of 1250° to 1450°C.

2.- The Cr-spinels allowed in the calibration has to be of Cr#<0.69, whereas at Los Hornitos, 7 of 11 spinels have Cr# above that limit and up to 0.76.

3.- Al is one of the slow diffusing elements that, together with Cr and P, can be incorporated into olivine during non-equilibrium growth in the form of boundary layers, and therefore, we are not confident of the geological meaning of the measured values.

Regarding the use of Ca in olivine to obtain temperatures (Köhler & Brey 1990): There is significant recent research regarding Ca partitioning. Recent work suggests that Ca in olivine is only weakly dependent on temperature (Feig et al., 2006). In fact, in arc settings Ca in olivine has been proposed to be quite sensitive to water contents in magmas and therefore may serve as a hygrometer (Gavrilenko et al., 2016). Therefore, we consider both of these methods to be not of great value to determine crystallization temperatures at Los Hornitos.

l.256- Give ascent rates in m/s for comparison with literature.

R: As previously mentioned, we have observed in the literature the use of both unit m/s and m/hr, and we have decided to add m/s for the entire manuscript in parentheses.

Figure S7- what do the dashed orange lines and white squares relate to? Are the Fo profiles from BSE or EPMA? Why do you see differences in rim compositions between the left and right sides of the mineral grain?

R: Thanks for this observation. We have improved the description of this figure in the caption to be clear with the mentioned uncertainties. The orange points are EMP spot analyses, the white squares are the BSE extracted gray values that were calibrated using the EMP spot analyses.

Regarding the difference in rim compositions: Please beware that the shown profiles in this figure (now Fig. S9) are excerpts of the full zoning profile. Rim compositions are not shown (see Fig. 2 for the full profile). The actual rim compositions agree quite well, given the steep zoning and therefore significant difficulties to obtain accurate rim compositions. The fact that there are small offsets between the left and right side in these profiles is not surprising. Please note that the vertical axis in each subfigure comprises about 1 Fo unit. Thus, actual compositional variations are only 0.2-0.4 Fo units, which we interpret to represent a variety of effects related to these natural samples: small variations in local melt chemistry, geometric effects, the imperfect growth of natural crystals, and potential small gradients in gray scale resulting from the actual image acquisition process. What is most important in our view here is the morphology of the zoning, which can be traced around each entire crystal. We do not expect identical composition but similar patterns in the crystal zones.

l.264- in between periods of rapid growth, or during periods of rapid growths?

R: Thank you for this observation. We have revised the writing and now we present a new paragraph to better explain the point.

Page 8-9: Line 283-291, new text:

“Linking textural and compositional data presented above allow us to determine that Ni reversals represent zones of rapid (diffusion-controlled) growth. Locally at the frame interface the rapid growth is followed immediately by transient slower growth zones represented by the Ni depletions, due to a decrease in the supersaturation that results from the partitioning of compatible elements into the growth zones (Ni, Mg) and the concurrent increase of incompatible elements. These chemical changes may induce a slow-down of crystal growth in the boundary layer environment (e.g., refs. ^{12,26,35,36}). While local boundary layers may slow down growth, dendritic fast growth of other new frames may continue simultaneously, documented in the advance of multiple frames (e.g., OL_32 and OL_50)”.

1.276- where is the evidence for these removed features? Without evidence this is just an assumption.

R: Recent publications have shown that dendritic and skeletal morphologies develop during diffusion-controlled growth regimes (Mourey and Shea 2019), and this implies the development of compositional boundary layers. These layers exert a first order control on the composition of olivine in these zones. We show in Figure S7 that in these domains, Ni experiences a larger impact compared to Fo, and therefore, we observe a greater depletion of Ni compared to forsterite in the olivines. In addition, the faster diffusion of Fo compared to Ni (Gordeychik et al., 2020), cause that Fo depletions may be erased quicker than Ni depletions. This can be observed in Fig. 2. For example, OL_37 have two main Ni depletions labeled as 1 and 2. Note that the inner Ni depletion is accompanied by a near absent Fo variation, whereas the outer Ni depletion is accompanied by a marked Fo depletion. And regarding your comment on line 204, we can estimate that these interior zones partially lose their zoning within weeks to months, leaving shoulders of flat zoning (with respect to Fo) behind. In conclusion, the evidence lies in the comparison of Fo gradients between the inner and outer Ni depletions.

We have included the citation of Gordeychik et al. (2020) at the end of these sentence, who has also looked at the timescales needed to erase such zoning, which is also on the order of weeks.

1.297- do you not see these elongations as the crystal continues to grow, and as you said evidence of any early growth is removed? So, you cannot comment on stage 1 per-se?

R: We are not clear what the reviewer is exactly referring to. In this sentence we are simply comparing our observations to the results of experiments of Mourey and Shea (2019), and we have not observed any olivine with elongation along the *a* axis, as they found in their experiments. However, others crystal features, as preferential growth along plane *ac* and tabular morphologies, are common in both investigations.

Section 5- It would be good to highlight what stage you attribute your timescales to within your crystal growth regime.

R: Thanks for this suggestion. We now include a short discussion about the relationship between calculated timescales and the growth model in the first paragraph of the “Discussion and Conclusion” section.

Page 11: Line 370-376, new text:

“Our calculated diffusion timescales are associated with the later stages of crystal maturation (~ Stage 4-5). These timescales cannot constrain the

individual episodes of rapid growth, nor do they provide information about the integrated time of the entire crystal growth. Since internal reversals are only documented in Ni reversals, while Fo zoning is preserved as flat shoulders, growth and residence may have been protracted for weeks to months allowing partial diffusive equilibration (ref. ²⁹). Alternatively, interior zones experienced less efficient CBLs, potentially due to slower overall growth in response to smaller degrees of undercooling¹⁵”.

1.333- again not sure you see evidence for the earliest growth stages.

R: It is clear that group 3 olivines represent the earliest stages of growth of olivine. Such morphologies have been previously observed at both experimental (Donaldson, 1976; Faure and Schiano, 2005) and natural samples (e.g. Colin et al., 2012; Shea et al., 2015). These natural samples are not rare crystals, and rather, represent an elusive growth stage, that given the short timescales of survival (days?) is commonly erased by overgrowth and maturation (see also Mourey and Shea 2019).

1.353- olivine thermometry will help test this theory of growth in mantle vs lower crust.

R: We have expanded the answer to this issue in comment to line 226. In short, olivine thermometry is not as definitive as the reviewer suggests. Also, as magmas move from the mantle to the crust, the local thermal regime depends on various aspects including the local geotherm, the size of the magma batch moving, the geometry of that batch, and the residence time. In addition, the diffusion profiles in the cores of olivine group 1, are a unique feature among the three groups, and the development of such diffusion profiles is expectable to occur in a nominally hotter environment (i.e. more efficient for diffusion) in the beginning of the growth history of those crystals.

1.368- how were these estimates of timescales calculated?

R: Thanks for this observation. These are calculations based in Fe-Mg interdiffusion. As previously pointed out, we have moved the comparison to Tassara et al. 2020 to an earlier part of the manuscript and removed it here.

Page 8: Line 260-269

1.369- do you mean > 102 days?

R: We made substantial changes to this part of the manuscript that included this comment; thus, it is not relevant anymore. The new paragraph that includes this part of the manuscript is now included in the “Timescales” section. See the answer to the previous major comment #2.

1.370- use consistent units for ascent rates

R: Thanks for this observation, we have corrected the units and now are reported in m/hr and m/s along the entire manuscript.

1.383- sentence needs rewording- unclear.

R: Thanks for this observation. We have improved this sentence.

Page 12: Line 422-424.

1.385- does overgrowth and diffusion within olivines really result in slower rates of magma ascent? I think this sentence needs flipping around.

**R: Thanks for this observation. We have revised this sentence to clarify our thinking.
Page 12: Line 425-427.**

1.388- Hildreth reference needed for MASH processes?

**R: Thanks for this observation. We have included the citation.
Page 12: Line 428.**

Fig. 2: Why are only mature olivines shown? Can you include an example of an immature olivine for comparison?

R: For the main purpose of the paper, which relies on textural features and chemical gradients along the different proposed structures of the crystals, the olivines of group 1 and group 2 provide the data to determine the Ni depletions and reversals that allowed to us the assessing of the conceptual growth model. On the other hand, immatures olivines of group 3 are not complexly zoned but provide key observations of the earliest stages of growth of natural olivine crystals, rarely preserved in arc magmas given the comparatively short timescales involved during their growth and the prolonged nominal residence times of basalt magmas.

To complement the data of group 3 olivines, we have included in Supplementary Fig. S4 a BSE and optical microscopy images of OL_106 and OL_67, respectively, with the location of EPMA measurements presented in Table S1. Additionally, please note that we also included two new data points for those olivines in Table S1. This is also in response to requests from Reviewer #2.

Response to Reviewer #2 Dr. Gerhard Wörner

Reviewer #2 (Remarks to the Author):

There are several things that I definitively like about this study:

This work presents evidence that concentric zoning in olivine crystals can be interpreted in a different way from what is usually implied. Standard interpretation is that Mg/Fe-reversals in olivine crystals (must) indicate magma mixing involving more or less evolved magmas. This interpretation is entrained into the current literature although several cases from hot-spot-related basaltic magmatism have been published, where zoning in slowly-diffusing clearly indicates otherwise.

This study presents a first case for such a process where skeletal growth is followed by textural maturation from an arc setting. As the authors state: “At Los Hornitos those stages of growth and maturation are preserved, in contrast to other arc volcanoes where they are commonly not observed.”

With such a new and contrasted view of the growth history, additional and potentially significant new insights may be possible when analysing and interpreting such zonation.

All analytical aspects of this study are fine, the documentation of the data, the textural interpretations, and the careful approach to diffusion modelling are convincing.

With respect to the P-estimates using Putirka’s calibration, I would suggest to be more careful and

include the error envelope into for the extracted depth-data in figure 5.

R: Thanks for this suggestion. We have incorporated in Fig. 5 the standard error of estimate (SEE) for each of the methods used to estimate pressure, and also, we have incorporated the pressure equivalence of depth into the cartoon to allow the reader evaluate the uncertainty in the depth calculations.

However, there are two serious issues that need to be addressed before this work can attain the level of becoming of interest to the readers of Nature Communications:

1. The method section mentions (and the supplement documents) analytical data for more slowly-diffusing elements (compared to Mg-Fe-Ni), namely Ca and Al. Unfortunately, it apparently was not possible to analyse Cr and P. Nevertheless, it is the hallmark of earlier publications that advocated the skeletal growth-to-textural maturation crystallisation process to document this process by zoning patterns of slowly-diffusing elements. Given that other examples from arcs have made exactly the opposite observation (concentric zonation of slowly-diffusing elements), it is a major problem in this study that such elements have not been analysed or not been used to make their case for early skeletal growth. Group 1 olivines in this study should show concentric Al-zonation and group 2 and 3 crystals should show “out-of-sequence” zonation for Al. This test has not been made here and this is really unfortunate.

R: We agree with reviewer: slow diffusing elements have been the hallmark of rapid growth regimes in olivine in many other publications.

As the reviewer pointed out, we collected compositional data of Al and Ca in olivines of group 1 (OL_39 and OL_49) and group 2 (OL_32). All these crystals present a broadly normal zoning of Al, with the exception of OL_49 that present a more complex zoning in the core. In addition, the Al zoning present major breaks (reversals?) along the profile (< 50 μm width) that consist of sharp higher contents that tend to occur flanking the borders of the core or either, coincide with the broad position of Ni reversals, that ultimately represent crystal frames in our model (see multielement compositional profiles in Table S1). This means that Al reversals may be linked to structures of rapid growth. On the other hand, Ca is monotonously constant along most of the profiles and reversely zoned in the outermost part of the crystals, where Fo evolve beyond Fo ~90. We consider these results interesting for further research but it remains unclear if any systematic variations exist for the case of Al, and thus we do not refer to these in the manuscript.

With regard of intensity maps of elements in our samples, we have included Figure S10 in the supplementary material showing olivines of different textural groups, for which we imaged major (Mg), minor (Ni) and trace elements (P). Cr and Al analysis does not provide enough resolution to resolve their distribution within the analyzed crystals. The selected olivines for this analysis are OL_38, an immature olivine of group 3, and OL_35, a mature olivine of group 2. As expected, the maps of Mg and Ni in these olivines reproduce the results obtained by compositional data obtained by EPMA, capturing the Ni depletions and reversals coupled to Mg variations. P occurs as concentric lines that remarkably tend to flank the structures of rapid growth (i.e. Ni reversals). This is expected, and consistent with a previous stage of increasing P contents in the chemical boundary layer.

In a broader sense, we consider the incorporation of slow diffusing elements by olivine as a secondary effect of rapid growth, where the Ni depletions and Fo-Ni reversals that we report, are the direct effect of rapid growth. As the slow diffusing elements require compositional boundary layers (CBL's) (Al) and rapid growth (P) to be incorporated by the growing crystal

(Shea et al., 2019), we can expect an alternated oscillatory zoning between high Fo-Ni zones (i.e. rapid-growth crystal frames of this study) and Ni depletions (developed under the effects of such CBL's), where the slow diffusing elements may be incorporated given the accumulation in the resultant CBL. If this is correct, then we can expect these elements to occur in the broad region marked by the Ni depletions of olivines reported here. A similar growth zoning was deduced by Shea et al. (2015), although they pointed out that Fo-Ni zoning may be difficult to be preserved or detected in natural samples, given the rapid diffusion of Mg-Fe, which our crystals partially confirm.

In essence, we propose that olivine growth proceeds via addition of crystal frames, which represent structures of rapid growth, forcing the emplacement of a chemical boundary layer enriched in incompatible components that may be incorporated into the growing olivine. Then, concentric oscillatory zoning of slow diffusing elements (P, Al, Cr) observed in intensity maps may not necessarily imply continuous concentric growth (i.e. tree ring model) as it may record the secondary effects of emplacement of such crystal frames in an -out of sequence-growth. Therefore, we disagree with the hypothesis raised by the reviewer since “concentric” and “out of sequence” zoning of these sluggish incompatible elements may be driven by a unique growth mechanism, here explained as concentric addition of crystal frames.

2. While I am prepared to accept the interpretation for the three-stage growth history of olivine crystals in this study, I wonder if not wider implications could be made. Is it that such detailed observations have not been made in basaltic arc magmas or is it that arc magmas indeed only rarely show such olivine growth histories. In the first instance, unfortunately, not much can be said. However, published and unpublished data indicate that such skeletal olivine growth may indeed be more common than previously thought in arcs, even on super-thick crust as in the Central Andes (e.g. Mattioli et al, 2006, J Volcanol Geotherm Res. 158: 87 – 105). If indeed skeletal growth is more typical in hot spot basalts, than in arcs, this difference must be explained. Maybe it is simply the different magma production rates and water contents that govern degassing and undercooling and the predominance of basalts in hot spots over more evolved compositions on arcs where primary mantle-derived basalts are occur more rarely.

In both cases, the question must be raised what the different growth histories may indicate with respect to different degrees of supercooling and how this may relate to systematically different magma potential temperatures, superheating of magmas, different water contents and effects of degassing and melt accumulation and ascent processes. All of these parameters will be different between hot spot basalts and arc basalts and a deeper consideration of such wider implications should be able to significantly enrich the present study. The manuscript touches on this issue in several instances but never goes to any depth.

R: Thanks for this major comment. We appreciate this request and think it is an important discussion to be added. While this discussion is only qualitative and somewhat speculative, it provides an opportunity to explore potential implications for fundamentally different crystallization and transport regimes that lead to the difference in OIBs and arc magmas. A more detailed discussion of this major implication has been incorporated to the manuscript in “Discussion and Conclusions” section (Page 12).

Page 12-14: Lines 432-483, new text:

“Ocean island settings have been the prevalent locations, where dendritic olivine growth has been described. The scarcity of studies describing dendritic and skeletal olivine growth in other settings raises the question,

whether this discrepancy is simply a sampling bias, i.e. studies have not looked for such olivines in sufficient detail, or is a result of fundamentally distinct magma storage and crystal growth regimes. Our understanding from experimental studies suggest that the formation and preservation of dendritic olivines requires rapid growth during times of large undercooling and no significant residence in the subvolcanic magma system to limit maturation and mechanical break up of dendritic and skeletal growth structures. The latter implies rapid ascent from regions of crystal growth to the surface where textures are quenched.

Large degrees of undercooling may emerge during magma mixing, interaction of small batches of hot magma with cool crust, or during decompression-driven degassing, which leads to sudden changes in the liquidus temperature. Magma temperature changes on the order of 40-60°C in response to mixing have been invoked^{15,48}. Thus, while mixing may be the driver for crystallization within hotter, more mafic magmas, very large temperature differences between the two mixing magmas may hinder the preservation of dendritic and skeletal olivine growth as such mixing is associated with more viscous intermediate to evolved magmas stored for extended periods in the crust. Run-up to eruption of such systems is commonly prolonged⁴⁹ likely leading to maturation of olivines and loss of dendritic features. Moreover, very large temperature gradients also cause abundant crystallization in the hot recharge magma and in arc settings this crystallization is dominated here by plagioclase associated with degassing and second boiling^{50,51}. The large changes in viscosity further slow-down any rapid transit to the surface and therefore the potential preservation of dendritic and skeletal olivines. Thus, magma mixing by itself may be a relatively inefficient process for dendritic olivine formation and preservation, especially in arcs. Rapid magma cooling in the crust is potentially a ubiquitous process and some studies have suggested cooling and differentiation during ascent¹⁰. However, in convergent margins with thicker crust and multiple crustal levels of magma storage^{1,45} uninterrupted ascent is fundamentally hindered, which may cause the lack of preserved dendritic and skeletal olivines in arcs, though exceptions exist and they are commonly limited to small volume batches⁵². Irrespective of whether such cooling of small batches during ascent is the cause for rapid crystal growth that may involve dendritic and skeletal textures, we speculate that such effects are important but not the lone cause for the abundant occurrence of dendritic and skeletal olivine in ocean island compared to arc settings. The fundamental difference between ocean island and arc settings is the volatile content dissolved prior to ascent. While ocean island basalts are commonly CO₂-rich and predominantly contain only small amounts of water, most arc magma volatile budgets are water-dominated. Thus, syn-eruptive degassing crystallization pathways are distinct. Compared to water, degassing of carbon dioxide starts deep and extends over a large pressure range. Effects on the phase stability and undercooling are thus prolonged and overall much smaller for carbon dioxide degassing^{53,54}. Thus, in arc settings with H₂O-rich but CO₂-poor magmas (i.e. low CO₂/H₂O)

undercooling accelerates during degassing and abundant crystallization may hinder eruption at the surface and leads frequently to stalling and textural maturation. Only magmas with high CO₂/H₂O ratios, that lead to a more continuous extraction of volatiles including H₂O during crustal ascent, have the ability to experience modest undercooling and avoid substantial crystallization and stalling in the shallow crust to eventually erupt at the surface. Therefore, we speculate in the potential absence of a sampling bias that volatile content may be the first order control that prompt the preservation of textures and multi-elemental concentric chemical zoning at olivines of Los Hornitos. The continuous CO₂ degassing provides the driving force for a sustained magma ascent from the deep crust that is associated with the moderate undercooling and allows these discrete primitive magma batches to reach the surface. This implies a close relation between primitive mafic monogenetic volcanoes, high CO₂ fluxing and the preservation of dendritic and skeletal textures in olivine. The future interrogation of volatile contents in olivine hosted melt inclusions will help to answer this hypothesis and it further highlights the need to reconstruct fully volatile budgets of magmas⁵⁵”.

So, in essence, I think that this study has great potential and eventually could become publishable in Nature Communications, but it clearly needs the consideration of wider context.

(BTW, it also needs a more catchy title...)

R: We agree, and have changed the title.

Gerhard Wörner

In addition to these comments from reviewer #2, we also made changes to the manuscript in response to his main comments within the pdf, listed below.

Line 46: “may be” followed, there are other examples

R: We made the change, even though we refer here to the findings and outcomes made by cited publications.

Page 2: Line 36.

Line 51: really different settings? All are oceanic hot spot

R: We note that many of the insights regarding olivine growth and complex zoning of slow diffusing elements comes from OIBs. However, there are other examples of such complex patterns in other settings as MORB, lunar basalts, arc basalts and even in equigranular triple-jointing olivine in dunite compositions associated to LIP's (Welsch et al., 2014; de Maisonneuve et al., 2016; Xing et al., 2017).

Line 57: sounds like this may be a wrong thing to do, but what when there are concentric growth bands for slowly- diffusing elements, then this implication may not be so wrong!

R: We are not arguing that all magmatic systems grow their olivines initially in the form of dendrites or skeletons, instead we want to challenge the opposite; the notion that all crystal are recording concentric in sequence growth. This assumption is not challenged in most

studies, and we think it should not be a fixed, definitive assumption. Thus, in this section we are describing the knowledge and achievements regarding the topic of this contribution without calling to any particular judgement. However, an important implication of this research is that concentric does not imply necessarily continuity (tree ring model). In fact, we determine that crystal frames grow concentrically but -out of core-to-rim sequence-.

Line 62: wording unclear, do you agree with our interpretation or not?

R: We are describing the broad scope and current knowledge and paradigm.

Line 349: hotter, yes, but why deeper?

R: This monogenetic cone does not have a long-lived magmatic system to cross. At least, there is no evidence for a very complex plumbing system underneath these cones that could disrupt the geotherm significantly. As a result, a typical general geotherm seems to us like a very reasonable assumption, with hotter conditions being also deeper.

Line 353: group 1 olivines have nucleated with small undercooling

R: Thanks for this suggestion. We have included it in the sentence.

Page 11: Line 392

Line 355: immature olivine shows maturation?? Just sounds strange!

R: Thanks for this observation. We have revised the sentence.

Page 11: Line 395

Fig.5: thin lines obscure uncertainty, change to wider band indicating errors of P-estimates

R: Thanks for this observation. We have included now the standard error of estimate (SEE) and a kbar scale to allow the reader to evaluate the uncertainties associated to the depth calculations into the cartoon.

Response to Reviewer #3 – Anonymous reviewer

This reviewer did not present line by line corrections in the pdf file and the pointed concerns are responded below.

Reviewer #3 (Remarks to the Author):

This article brings together information from both textures and diffusion patterns of olivine phenocrysts from a monogenetic centre in Southern Andes. From the different stages of maturation and compositional reversals in those crystals the authors propose an evolutive pattern that is more complex than expected and notably preserves early stages of olivine nucleation. The latter is significant because upper crust differentiation of arc basalts usually deletes the early structure as crystals interact with impending magmas or they just stall in low pressure conditions. Taken into account this complexity the authors identify diffusive zones and use them to estimate timescales and ultimately a very fast ascent through the crust. The authors make a good selection of the case-study, take a novel approach combining detailed textural analysis and compositional gradients and thus shed light on a usually elusive stage of crystal growth. The short residence time is somewhat expected in mafic magmas from minor centres but only quantitatively estimated in a few cases yet.

Some specific questions/comments are listed below:

(L78). Although beyond the scope of this article, ‘structural control’ itself seems to be not enough to explain mafic magmas from Los Hornitos crossing the crust without interruption. Quoted references in fact show other monogenetic centres under a similar ‘structural control’, but slightly more evolved. Is there any other potential reason for such a primitive signature?. Can be the other mafic magmas from monogenetic centres modelled with a common mantle source and melting conditions?

R: The reviewer refers to the citation of Salas et al. (2016), and the mentioned “slightly more evolved mafic cones” distribute towards the eastern flank of the arc, whereas Los Hornitos locates in the axis of the arc. In fact, there is a significant difference among these two domains with respect to the basement that the feeding magma systems have to traverse in their way to reach the surface. The volcanoes distributed to the east have to traverse a comparatively more complex stratigraphic rock column, Eastward the Miocene tuff sequence known as Campanario Formation (Drake, 1976), may act as an impeding rock layer where magmas are trapped and forced to evolve (and likely assimilate chemically evolved material), that lead to more evolved compositions emplaced on the surface. By contrast, Los Hornitos are located out of the domain of the Campanario Formation, but in addition, their location coincides with a major regional contact between the Abanico Formation (Oligocene; Aguirre, 1960) with the Mesozoic sequences of Baños del Flaco Formation (Early Cretaceous; Khlon, 1960) and the Miocene granitoids that constitute the basement of the current volcanic front in the area (Fig. 3 and 4 in Salas et al. 2016). Therefore, a first order factor controlling the location of Los Hornitos is given by a “structural control” of regional scale but also, the more evolved character of mafic vents distributed to the east may be explained by more prolonged residence times and rock assimilation in the thick crust.

In conclusion, the existence of a thicker Cenozoic basement to cross to achieve the surface in the eastern mafic vents may be a potential reason controlling the more evolved character of these vents instead of different magma sources or melting conditions.

This topic is currently part of a work in preparation by an undergraduated student, and preliminar results were shown at the LASI Conference 2019 (Espinosa-Leal and Salas, 2019). See attached figure below.

[Redacted]

(L86). The study focuses on the tephra sequence related to the younger vent, where magnesian olivines dominate, but it would be good to know more about the sequence associated to the older cone. The latter is the first dike ascending so a comparison could inform on relative ascent velocities, which could be higher for the second if the channel was already established. A direct test in these two sequences from the same vent (but probably close in time) would be an additional demonstration of the growth model proposed (with other constraints remaining the same).

R: We agree with the reviewer that analysing both eruptive events would lead to a more comprehensive understanding of these two eruptive events, however, tephra deposits related with the earlier stages of the older cone are significantly less accessible. Two chemical analyses in these tephra samples, indicate similar patterns in major and minor elements compared to the tephra of the younger cone. We prefer to use the well-preserved tephra deposits associated to the younger cone where sample correlation to the fallout column is much clearer. Reviewer #1 has requested more clarity with regard the time relations and distribution of volcanic products associated with the two cones in their comment to line 92, where more details regarding to these concerns can be found. In addition, a new figure in the supplementary material was included to clarify the setting (Fig. S1). We consider that the decision of using the tephra samples associated to the younger cone does not invalidate the proposed growth model and its related implications.

(L237). What is the actual Fo range in the core to rim sequence?. What the meaning of the outermost rims that evolve beyond Fo88?.

R: As the different olivine groups were nucleated sequentially during source magma evolution, the specific zoning patterns are not the same among the groups. Excluding the outermost rim, olivines of group 1 range between ~Fo92 in the innermost core to ~Fo90.5 toward the rim; whereas group 2 ranges between ~Fo91.5 to ~Fo89.5 in the innermost core and toward the rim, respectively. Finally, olivines of group 3 have a comparatively smoother Fo zoning, with broad values of Fo91.3 to Fo90, as the case of OL_106 (supplementary material, Table S1 and Fig. S4).

Regarding the meaning of the outermost rims that evolve beyond Fo88, we consider them to result of the more pronounced cooling stage at scale of the subvolcanic conduit, during the co-crystallization of olivine (Fo<~88) in addition to clinopyroxene and plagioclase previous to cooling in the surface.

(L250). What is implied in physical terms from the difference between growth of crystals from Kilauea and those from Los Hornitos?. Magma ascent is thought to be fast in monogenetic centres (and thus has been estimated in this article) and hence no significant difference would be expected with basalts from oceanic islands, especially for the rejuvenated stage. Could be this Peclet number telling something about the early crystallization stage before immediately the ascent?

R: Thanks for this comment, we agree with the point of view of the reviewer.

We think these concerns are similar to the point raised by reviewer #2 and we discuss this in response to his main comment #2.

Regarding the Peclet number argument we present, we are not sure what the reviewer is suggesting. In general, this argument simply addresses the point that we commonly assume that growth is faster than diffusion and step functions may be a useful approach when modelling diffusion. While we also do this here, it clearly represents a maximum timescales assumption if growth and diffusion actually operate on similar timescales (i.e. the Peclet number may be close to 1).

References

- Aguirre, L. 1960. Geología de los Andes de Chile central, Provincia de Aconcagua. Instituto de Investigaciones Geológicas. Boletín 9: 70 pp. Santiago.
- Albert, H., Larrea, P., Costa, F., Widom, E., Siebe, C., 2020. Crystals reveal magma convection and melt transport in dyke-fed eruptions. *Scientific Reports* 10, 11632
- Colin, A., Faure, F., & Burnard, P. (2012). Timescales of convection in magma chambers below the Mid-Atlantic ridge from melt inclusions investigations. *Contributions to Mineralogy and Petrology*, 164(4), 677-691.
- Coogan, L. A., Saunders, A. D., & Wilson, R. N. (2014). Aluminum-in-olivine thermometry of primitive basalts: Evidence of an anomalously hot mantle source for large igneous provinces. *Chemical Geology*, 368, 1-10.
- de Maisonneuve, C. B., Costa, F., Huber, C., Vonlanthen, P., Bachmann, O., & Dungan, M. A. (2016). How do olivines record magmatic events? Insights from major and trace element zoning. *Contributions to Mineralogy and Petrology*, 171(6), 1-20.
- Donaldson, C. H. (1976). An experimental investigation of olivine morphology. *Contributions to Mineralogy and Petrology*, 57(2), 187-213.
- Drake, R. E. (1976). Chronology of Cenozoic igneous and tectonic events in the central Chilean Andes—Latitudes 35° 30' to 36° S. *Journal of Volcanology and Geothermal Research*, 1(3), 265-284.
- Faure, F., & Schiano, P. (2005). Experimental investigation of equilibration conditions during forsterite growth and melt inclusion formation. *Earth and Planetary Science Letters*, 236(3-4), 882-898.
- Feig, S. T., Koepke, J., & Snow, J. E. (2006). Effect of water on tholeiitic basalt phase equilibria: an experimental study under oxidizing conditions. *Contributions to Mineralogy and Petrology*, 152(5), 611-638.
- Gavrilenko, M., Herzberg, C., Vidito, C., Carr, M. J., Tenner, T., & Ozerov, A. (2016). A calcium-in-olivine geohygrometer and its application to subduction zone magmatism. *Journal of Petrology*, 57(9), 1811-1832.
- Gordeychik, B., Churikova, T., Shea, T., Kronz, A., Simakin, A., & Wörner, G. (2020). Fo and Ni Relations in Olivine Differentiate between Crystallization and Diffusion Trends. *Journal of Petrology*.
- Klohn, G. 1960. Geología de Santiago, O'Higgins, Colchagua y Curicó. Instituto de Investigaciones Geológicas Chile. Boletín 8: 95 pp. Santiago.
- Köhler, T. P., & Brey, G. (1990). Calcium exchange between olivine and clinopyroxene calibrated as a geothermobarometer for natural peridotites from 2 to 60 kb with applications. *Geochimica et Cosmochimica Acta*, 54(9), 2375-2388.

- Milman-Barris, M. S., Beckett, J. R., Baker, M. B., Hofmann, A. E., Morgan, Z., Crowley, M. R., ... & Stolper, E. (2008). Zoning of phosphorus in igneous olivine. *Contributions to Mineralogy and Petrology*, 155(6), 739-765.
- Mourey, A. J., & Shea, T. (2019). Forming olivine phenocrysts in basalt: a 3D characterization of growth rates in laboratory experiments. *Frontiers in Earth Science*, 7, 300.
- Ruth, D. C., Costa, F., de Maisonneuve, C. B., Franco, L., Cortés, J. A., & Calder, E. S. (2018). Crystal and melt inclusion timescales reveal the evolution of magma migration before eruption. *Nature communications*, 9(1), 1-9.
- Salas, P. A., Rabbia, O. M., Hernández, L. B., & Ruprecht, P. (2017). Mafic monogenetic vents at the Descabezado Grande volcanic field (35.5 S–70.8 W): the northernmost evidence of regional primitive volcanism in the Southern Volcanic Zone of Chile. *International Journal of Earth Sciences*, 106(3), 1107-1121.
- Shea, T., Lynn, K. J., & Garcia, M. O. (2015). Cracking the olivine zoning code: Distinguishing between crystal growth and diffusion. *Geology*, 43(10), 935-938.
- Shea, T., Hammer, J. E., Hellebrand, E., Mourey, A. J., Costa, F., First, E. C., ... & Melnik, O. (2019). Phosphorus and aluminum zoning in olivine: contrasting behavior of two nominally incompatible trace elements. *Contributions to Mineralogy and Petrology*, 174(10), 85.
- Tassara, S., Reich, M., Cannatelli, C., Konecke, B. A., Kausel, D., Morata, D., ... & Leisen, M. (2020). Post-melting oxidation of highly primitive basalts from the southern Andes. *Geochimica et Cosmochimica Acta*, 273, 291-312.
- Welsch, B., Hammer, J., & Hellebrand, E. (2014). Phosphorus zoning reveals dendritic architecture of olivine. *Geology*, 42(10), 867-870.
- Xing, C. M., Wang, C. Y., & Tan, W. (2017). Disequilibrium growth of olivine in mafic magmas revealed by phosphorus zoning patterns of olivine from mafic–ultramafic intrusions. *Earth and Planetary Science Letters*, 479, 108-119.
- Xing, C. M., Wang, C. Y., & Tan, W. (2017). Disequilibrium growth of olivine in mafic magmas revealed by phosphorus zoning patterns of olivine from mafic–ultramafic intrusions. *Earth and Planetary Science Letters*, 479, 108-119.

REVIEWERS' COMMENTS

Reviewer #1 (Remarks to the Author):

The authors have made significant improvements to the manuscript and have addressed key comments from both mine and others' previous reviews. The paper reads well, and I have just a few minor comments, which I have included on the tracked changes version of the manuscript that the authors uploaded. Some of these are suggested rephrasings (which the authors may chose to ignore) others are spelling/grammar corrections, and finally some are comments where slight adjustments may add to the overall clarity of the manuscript.

I have one slightly more involved comment, involving the final conclusion that the ratio of CO₂ to H₂O is an important control in the development of dendritic morphologies in arc environments. In the methods section, the authors give details of SIMS analyses of glass selvages for volatile concentrations, yet this data is not presented anywhere. If the authors have volatile data then this should be integrated into the manuscript to test their final conclusion.

I am happy to recommend this manuscript for publication following minor modifications,

Kind regards
Katy Chamberlain

I have assessed all comments by the authors that respond to all three reviewers remarks. In all, I find that they have responded well and stated their case convincingly.

I am now even more supportive of publication of this work. It is provocative in a positive sense in that it presents a fundamentally different approach to the interpretation of compositional zonation in olivine.

Why is this important ?

Olivine compositional zonation have developed as the major tool to unravel the history of basaltic melts from deep magma processing, to ascent and final eruption. Many studies have interpreted such compositional zonation, often based on diffusion modelling to extract rates of ascent, duration of magma residence and timing and process of magma mixing. These are topics at the forefront of igneous petrology, which are not without implications for eruption forecasting and hazard mitigation.

Their new view that concentric compositional zonation in olivine may develop through out-of-sequence growth of crystal frames could have a major impact on how to interpret zonations and their diffusion timing.

They have chosen a (rather rare ?) case of a near-primitive basalt erupted in a continental arc setting and argue that growth via olivine frames before textural maturation may be more common and that previous interpretations of concentric growth textures may be in error.

However, while being slightly bold about their new interpretation is fine, and in order for it to have corresponding impact, it requires tight(er) and (more) convincing arguments.

The authors have offered three arguments, that I will address below to support my recommendation to provide their response and deeper discussion. This may be an anticipation of the discussion that could follow after eventual publication and therefore, this might be a worthwhile exercise rather nit-picking.

Abstract

I have taken the liberty to slightly modify the abstract to make it more concise and straight(er) to the point. It may be helpful to catch the readers interest right from the start.

Do concentric P zonation patterns indicate skeletal growth or not ?

It may not be a sufficient argument to consider concentric P-oscillations to always indicate skeletal growth, and that it may be equally wrong to assume that it must always indicate concentric polyhedral growth. So far, I agree.

However, I would argue that only zonation patterns that are actually skeletal, such as those shown by the frequently (and justly) cited Welsch et al (2014, 2014) references, are convincing arguments. We did not see such clear evidence in the Shiveluch examples such as presented by Welsch et al (2013, 2014). I also agree that not all their patterns are skeletal and some are in fact quite concentric and look sequential but may not be so, according to these authors. It is true, we did not (even) consider skeletal growth, in fact, we never even use the terms "skeletal" or "dendritic" anywhere in our publication. Therefore, we did not "*specifically exclude skeletal growth*". It is only implicit, that we did

not consider such interpretation. This is really a minor point, but I suggest to get this correct.

In any case, we all agree that such oscillatory P-zonation patterns are the boundary layer effects, as we (Gordeychik et al., 2018) wrote : „we ascribe the narrow oscillatory growth bands ... (of P and Al) ... to kinetic effects rather than growth from changing melt compositions“.

These comments should not be taken necessarily as a defence of the Gordeychik et al (2018) interpretation, that is not my point (our interpretation may well be wrong). But I am still puzzled by the *implicit* assumption that any concentric P + Al zonation in olivine somehow “must always” be evidence for skeletal growth. I am not even sure if the authors really intend to say that, but the text appears to indicate such *inclination*. This impression arises when the text says (in short): “concentric P-zonation is not a sufficient argument for skeletal growth (fine !) but there are three additional arguments that indeed they must be”. Here, I do not agree!

The authors should clear up where they stand, which is just not quite clear: Does concentric oscillatory zonation in P and Al, which is clearly kinetically controlled, (always) indicate skeletal/dendritic growth or not?

There is also a crucial observation that may help. Skeletal growth is beautifully shown in this study and it consists of frames that are jointed by some sort of a radiating array of connectors (which is a necessary non-concentric element of the growth frame texture). After subsequent maturation of crystal frames the connectors should still exist. Welsch et al (2013, 2014) actually show traces of non-concentric by P zonation that document the prior skeletal shapes. I admit that this is not true for all Welsch et al images and still they would argue (as the present authors would probably also do) that those crystal with only concentric P zonation patterns are also matured skeletal crystals. However, in that case the traces of radial connectors should remain visible in P-zonation, which has not been shown here (This was my point when requesting to see the zonation patterns of slow-diffusing elements in my earlier review). So, why do we see the frames in P-zonation but the connectors we do not see ?

Again, to make sure, I am not at all against the interpretation that skeletal crystals can texturally mature into polyhedral crystals. Actually, I think this is, among others, a very normal process of olivine crystallisation in certain settings. However, my conclusion from all of this is that, possibly, initial concentric polyhedral growth AND maturation of crystal frames into polyhedral shapes could both occur. If so, these two growth mechanisms could occur in different lavas but also within the same lava if the thermal history of its batches is complex (more on that below). With all that, the three major arguments that magma mixing is not a likely process, becomes rather weak.

Authors additional arguments in support of their crystal framing model, based on concentric P-zonation patterns

Later in the manuscript, the authors present observations to support their interpretation, and argue against magma mixing. These arguments are:

1. Mixing would be just too a complex process, in particular given the short time allowed
2. Fo-levels of different growth zones are all compositionally different
3. Different crystal types nucleated in sequence and thus represent different stages of crystallisation and thus age of crystals.

These arguments are not fully conclusive to me, for several reasons:

The authors argue that oscillating Fo during a short period of ascent (days) indicates just too a complex plumbing history with rapidly changing melt compositions. I follow the first argument of the authors that such extremely rapid mixing in small dikes would be unlikely for large compositional contrasts between magmas, and for more evolved magmas, but that is not the case here. Repeated mixing, cooling, crystallisation, and again mixing could, in fact, be expected during fast ascent as long as the observed compositional contrast between mixing magmas is small. IN a dike, different magma domains could form and mix in strong temperature gradients, even convective motion in dikes has been proposed. This would allow for different magma batches, all quite mafic, with different types of crystals. These could the mix and co-erupt.

In fact, the authors also explain the oscillatory zonation of P to be due to kinetic effects in a boundary layer. This interpretation must imply rapidly changing growth rates, which then implies rapidly changing temperature conditions. This is unless constitutional undercooling is implied here, which is unlikely because small contents of Al and P do not affect the olivine solidus temperature of the melt. If the kinetic effect is due to growth rate (and thus bulk temperature) changes, then the thermal history of the melt must be highly variable at short time intervals. Conductive processes within the bulk of the melt should then be ruled out and this only leaves (in my opinion) mixing between differently cooled melt batches that form during rapid (even turbulent ??) flow during ascent through a narrow conduit. This may be accompanied by flow differentiation to cause minor changes in melt compositions of mixing magma batches.

These melt portions also would all be slightly different, which would explain the different Fo-contents in the different “frames” (authors second argument). The third argument by the authors is that all crystal types occur in all parts of the tephra deposit even though they nucleated sequentially. In my opinion, this is not at all in conflict with the fast mixing in a conduit. In such scenario olivine crystals would indeed be forming all the time from the different melt batches so crystals from all times and stages of nucleation, skeletal growth and maturation would form and be erupted together.

I still can agree with the authors that in their case, magma mixing may not necessarily be the fundamental process that results in concentric zonation. So their conclusions that “The presence of multiple magma batches stored in the crust to account for several compositional reversals recorded in olivine phenocrysts are not required” may still be justified. But I would like to see additional arguments !

If their case can be a bit better argued, then this example will firmly establish an alternative interpretation, break the standard assumptions about concentric growth

caused by magma mixing and should be always accepted as an alternative explanation for concentric zonation.

A minor fine-tuning of the text here and there to address these issues should be easily done and not require too much effort, but it would be worth doing it.

Wider implications

In my review I suggested to explore the wider implications of this study. This concerns the question whether arc magmas, due to their higher water contents, their different ascent and magma plumbing system in different crustal stress regimes may – or may not – systematically be different with respect to the thermal history and olivine crystallization during the passage of basalt magma from source to surface.

I am now more than happy with the extended discussion, this has been done extremely well.

Again, none of these comments should be taken to suggest that this work should not be accepted to Nat Comm. Additional improvements will make this an important contribution to our understanding of ascent and crystallisation of basaltic arc magmas and their potential contrast to other settings.

25.1. 2021

Gerhard Wörner

Reviewer #3 (Remarks to the Author):

This version expands the discussion on the importance of skeletal growth in arc settings compared to others, primarily ocean islands. Although speculative, the authors mention potential causes for such abundance of textures in oceanic basalts, and why they are present in the case-study. This addition helps in making the article attractive to a wider audience.

The study focuses on the tephra sequence related to a younger vent, where magnesian olivines dominate and textures under study are visible. Although access to the early pyroclastic sequence is presumably hard, some inferences about the ascent velocities would be stronger with a comprehensive study of the entire sequence. Being a scarce and apparently elusive texture in arc settings, another example from a close related scenario (same geochemical features and structural setting, as expected for the older cone) would be useful to confirm this behavior shedding light on the mechanisms driving this growing history. However, the latter in fact does not invalidate the conclusions about the growth model and all related implication, which would be stronger with a paired case.

Responses to reviewers #2

Reviewer #1 Dr. Katy Chamberlain

We are very thankful of the new comments and text edits to our revised manuscript. We have accepted most of them. In regard to the CO₂ & H₂O requested by the reviewer; unfortunately we are not fully confident in the CO₂ data, since our reference glasses contained CO₂ in one to two orders of magnitude smaller than the obtained values and the reference glasses had high CO₂ backgrounds, likely from contamination during the preparation. Therefore, we prefer not publishing these data. We have included a new table (Table S5) to show the composition in H₂O, S, Cl, P and F obtained by SIMS on the external glasses surrounding olivine crystals. The averaged H₂O content measured in these glasses ($n=20$) was used into the barometry calculations. Note in the "Methods" section that we have corrected the previous reported number of spots from $n=22$ to $n=20$ and also we have cited the new table (Table S5).

Reviewer #2 Dr. Gerhard Wörner

We appreciate the opportunity to clarify some of the concepts in our paper and provide additional arguments why mixing is not a satisfactory interpretation. We also would like to acknowledge that their recent paper on Shiveluch (Gordeychik et al. 2018) provides an intriguing useful study against which to compare our results. We think there are fundamental differences in the Shiveluch study and our finding and thus, we restrain ourselves from reinterpreting their results.

See below our responses into the quoted text of the reviewer:

I have assessed all comments by the authors that respond to all three reviewers remarks. In all, I find that they have responded well and stated their case convincingly.

I am now even more supportive of publication of this work. It is provocative in a positive sense in that it presents a fundamentally different approach to the interpretation of compositional zonation in olivine.

Why is this important ?

Olivine compositional zonation have developed as the major tool to unravel the history of basaltic melts from deep magma processing, to ascent and final eruption. Many studies have interpreted such compositional zonation, often based on diffusion modelling to extract rates of ascent, duration of magma residence and timing and process of magma mixing. These are topics at the forefront of igneous petrology, which are not without implications for eruption forecasting and hazard mitigation.

Their new view that concentric compositional zonation in olivine may develop through out-of-sequence growth of crystal frames could have a major impact on how to interpret zonations and their diffusion timing.

They have chosen a (rather rare ?) case of a near-primitive basalt erupted in a continental arc setting and argue that growth via olivine frames before textural

maturation may be more common and that previous interpretations of concentric growth textures may be in error.

R: We don't argue that concentric growth is fundamentally the wrong assumption. There remains a continued need for experiments and studies from natural examples to fully grasp crystal growth, especially in the context of olivine.

Instead, we see our paper as an alternative interpretation to the concentric growth paradigm and we are in general agreement with observations presented in Welsch et al. 2013 (for an OIB system). As the reviewer highlighted, these findings will urge the community to consider skeletal growth as a viable option for olivine growth and diffusion studies. In our opinion, such an alternative has to be considered moving forward with the mounting evidence from textural work for the potential skeletal growth of olivine as a common mechanism, including this manuscript.

However, while being slightly bold about their new interpretation is fine, and in order for it to have corresponding impact, it requires tight(er) and (more) convincing arguments.

The authors have offered three arguments, that I will address below to support my recommendation to provide their response and deeper discussion. This may be an anticipation of the discussion that could follow after eventual publication and therefore, this might be a worthwhile exercise rather nit-picking.

Abstract

I have taken the liberty to slightly modify the abstract to make it more concise and straight(er) to the point. It may be helpful to catch the readers interest right from the start.

R: Thanks for these suggestions. We have considered most of them.

Do concentric P zonation patterns indicate skeletal growth or not ?

It may not be a sufficient argument to consider concentric P-oscillations to always indicate skeletal growth, and that it may be equally wrong to assume that it must always indicate concentric polyhedral growth. So far, I agree.

However, I would argue that only zonation patterns that are actually skeletal, such as those shown by the frequently (and justly) cited Welsch et al (2014, 2014) references, are convincing arguments. We did not see such clear evidence in the Shiveluch examples such as presented by Welsch et al (2013, 2014).

I also agree that not all their patterns are skeletal and some are in fact quite concentric and look sequential but may not be so, according to these authors. It is true, we did not (even) consider skeletal growth, in fact, we never even use the terms "skeletal" or "dendritic" anywhere in our publication. Therefore, we did not "*specifically exclude skeletal growth*". It is only implicit, that we did not consider such interpretation. This is really a minor point, but I suggest to get this correct.

R: We agree that we misrepresented the Gordeychik et al. (2018) study. The authors did not explicitly reject a "skeletal" growth model. In fact, they did not cite the relevant papers for that model (e.g., Welsch 2013, 2014) and therefore

they did not consider such a growth model. Thus, we have modified the manuscript text in line 57 accordingly.

In any case, we all agree that such oscillatory P-zonation patterns are the boundary layer effects, as we (Gordeychick et al., 2018) wrote : „*we ascribe the narrow oscillatory growth bands ... (of P and Al) ... to kinetic effects rather than growth from changing melt compositions*“.

These comments should not be taken necessarily as a defence of the Gordeychik et al (2018) interpretation, that is not my point (our interpretation may well be wrong). But I am still puzzled by the *implicit* assumption that any concentric P + Al zonation in olivine somehow “must always” be evidence for skeletal growth. I am not even sure if the authors really intend to say that, but the text appears to indicate such *inclination*.

R: We do not intend to state that all concentric P + Al zoning is unconditional evidence for skeletal growth, and neither have we made that statement; neither explicitly, nor implicitly.

We only address P zoning briefly in l. 308-313 for Los Hornitos for two immature crystals. As we stated before we don't have complete P-maps for polyhedral crystals and therefore do not know whether P-zoning concentric around the entire crystal exists. We should note that perfectly concentric P-zoning for an entire crystal has not really been shown in other studies either (Shea et al 2015; Millman-Barris et al 2008; Welsch et al 2013): concentric zoning is commonly incomplete. This is one other argument why we don't state that all concentric P + Al zoning is unconditional evidence for skeletal growth. It is not even clear whether complete (360°) concentric P + Al zoning exists. Corners are always enhanced in all examples.

We then go on to highlight that the P-zoning we do see correlates with the observed Ni depletions (l. 309) and textural arguments. Therefore, in the case of Los Hornitos we would predict that skeletal growth can create almost complete concentric P-zoning. Thus, we state that the observed concentric Fo-Ni zoning in Los Hornitos is consistent with an out-of-sequence growth model. And we expand this concept to concentric oscillatory Fo-Ni zoning in olivine in general. I.e., we state that concentric zoning does not imply necessarily core to rim sequential growth!

We carefully word this as an alternative to sequential growth that ought to be considered moving forward. In our view, the textural evidence from hundreds of crystals is highly supportive for Los Hornitos; in other systems, a case has to be made for either sequential or out-of-sequence growth (we think that such cases can be made – independent of P + Al maps if the textures are supportive).

This impression arises when the text says (in short): “*concentric P-zonation is not a sufficient argument for skeletal growth (fine !) but there are three additional arguments that indeed they must be*”. Here, I do not agree!

R: We think that the reviewer is adding his own interpretation of our text, by paraphrasing incorrectly or insufficiently. Given that there is no line number reference here, it is unclear where this text originated that is in quotation marks. It is not a quotation, but paraphrased and therefore, we have to disagree with this representation. We think skeletal growth at Los Hornitos is

well evidenced (as agreed by the reviewer) and concentric zonation (Fo-Ni and/or P) can be consistent with this growth regime. And turned around: concentric zonation alone is not sufficient evidence to argue exclusively for sequential core-to-rim growth.

The authors should clear up where they stand, which is just not quite clear: Does concentric oscillatory zonation in P and Al, which is clearly kinetically controlled, (always) indicate skeletal/dendritic growth or not?

R: We disagree with the reviewer's position here to force an unconditional statement. We think for Los Hornitos the case is clear, but our intention with this manuscript is to start new community research to test more broadly when concentric zoning may represent sequential growth and when it represents out-of-sequence growth. The fact that at Los Hornitos out-of-sequence growth is convincing (as noted by the reviewer) raises the question under what conditions this occurs. Experimental data suggest that undercooling controls the textural evolution.

There is also a crucial observation that may help. Skeletal growth is beautifully shown in this study and it consists of frames that are jointed by some sort of a radiating array of connectors (which is a necessary non-concentric element of the growth frame texture). After subsequent maturation of crystal frames the connectors should still exist. Welsch et al (2013, 2014) actually show traces of non-concentric by P zonation that document the prior skeletal shapes. I admit that this is not true for all Welsch et al images and still they would argue (as the present authors would probably also do) that those crystal with only concentric P zonation patterns are also matured skeletal crystals. However, in that case the traces of radial connectors should remain visible in P-zonation, which has not been shown here (This was my point when requesting to see the zonation patterns of slow-diffusing elements in my earlier review). So, why do we see the frames in P-zonation but the connectors we do not see ?

R: Radial connectors are referred throughout the manuscript as dendritic extensions (described in detail at I. 98-104) and correspond to a planar feature that only occurs along the crystallographic plane ac . Therefore, these radial connectors are not visible in other crystal sections and neither are they easily exposed so well that P-maps can be readily collected. Unfortunately, we did not analyze the ac plane in the samples imaged by X-rays mapping – a shortcoming that we cannot fix at this point. However, as for Los Hornitos, the exemplary crystal of Shea et al 2015, consist of a skeletal olivine oriented along the ac plane and was mapped by X-rays. There it was shown that these

radial connectors are conformed by high P intensities. We expect similar P-zoning patterns in the olivines of Los Hornitos along the ac plane. We intend to add more X-rays maps of new crystals of Los Hornitos in the future in follow up work.

Again, to make sure, I am not at all against the interpretation that skeletal crystals can texturally mature into polyhedral crystals. Actually, I think this is, among others, a very normal process of olivine crystallisation in certain settings. However, my conclusion from all of this is that, possibly, initial concentric polyhedral growth AND maturation of crystal frames into polyhedral shapes could both occur.

R: We agree with that statement. But we do think that in the case of Los Hornitos (and possibly other volcanoes) it is better explained by skeletal and subsequent maturation to form polyhedral olivines!

If so, these two growth mechanisms could occur in different lavas but also within the same lava if the thermal history of its batches is complex (more on that below). With all that, the three major arguments that magma mixing is not a likely process, becomes rather weak.

R: We think mixing can be a very important process in arc magmas. Co-author Ruprecht is a major proponent for open system behaviour and has published on that on several occasions with respect to both the fluid dynamics and the geochemical record. Thus, other systems may very well be affected by mixing and such mixing of course can create reverse zoning and potentially even oscillatory zoning. However, the case at Los Hornitos is a poor choice to argue for mixing.

We now extended our arguments why mixing is not a convincing process here and added a brief discussion regarding to this point in lines 185-202, citing other examples of compositional variations in olivine induced by magma mixing (see also next discussion point):

Page 6, l.185-202:

“The standard interpretation of the olivine zoning at Los Hornitos would commonly invoke magma mixing, however we propose the out-of-sequence growth model as an alternative interpretation for the following reasons. Compositional zoning is uniform among the olivine grains with a uni-directional evolution to more evolved compositions at the rim interrupted by short wavelength Ni depletions. Fluid motion during mixing in magmatic systems is mostly chaotic, especially in reservoirs, but potentially also in conduits. Thus, crystals nucleating and growing in a mixing, rapidly evolving environment display a great variety of compositional histories (e.g., refs. ^{28,29}). Some crystals may originate in the more evolved melt whereas others in the original primitive melt, leading to complex crystal populations. Moreover, during mixing, crystals may reside for different time scales in the compositionally distinct melts leading to variable lengthscales of zonation. Both of these characteristics are inconsistent with the uniform and uni-directional evolution we observe. If mixing is not only chaotic, but approaches turbulent conditions, local compositional gradients would be quickly erased producing very simple zoned crystals.

By comparing the Los Hornitos olivine zoning to other strongly-zoned olivines that are interpreted to represent magma mixing, we find important differences. The magnitudes of Fo-Ni increments associated with the growth of crystal frames ($\Delta\text{Fo} < 0.6$ and $\Delta\text{Ni} > 200$ ppm) contrast with Fo-Ni variations reported for a ring tuff of Shiveluch volcano in Kamchatka¹⁸, Llaima volcano¹⁷ in the Andean Southern Volcanic Zone of Chile and Parícutín in Central America³⁰.

Authors additional arguments in support of their crystal framing model, based on concentric P-zonation patterns

Later in the manuscript, the authors present observations to support their interpretation, and argue against magma mixing. These arguments are:

1. Mixing would be just too a complex process, in particular given the short time allowed
2. Fo-levels of different growth zones are all compositionally different
3. Different crystal types nucleated in sequence and thus represent different stages of crystallisation and thus age of crystals.

These arguments are not fully conclusive to me, for several reasons:

The authors argue that oscillating Fo during a short period of ascent (days) indicates just too a complex plumbing history with rapidly changing melt compositions. I follow the first argument of the authors that such extremely rapid mixing in small dikes would be unlikely for large compositional contrasts between magmas, and for more evolved magmas, but that is not the case here. Repeated mixing, cooling, crystallisation, and again mixing could, in fact, be expected during fast ascent as long as the observed compositional contrast between mixing magmas is small. IN a dike, different magma domains could form and mix in strong temperature gradients, even convective motion in dikes has been proposed. This would allow for different magma batches, all quite mafic, with different types of crystals. These could the mix and co-erupt.

R: This reasoning is highly speculative. And we strongly disagree for the following reasons. 1) Zoning patterns of different crystals (while slightly different in their concentrations) follow the same sequence: Broad normal zoning is disrupted with short intervals of reverse zoning. Each and every crystal shows this! Thus, there is a directionality in the data. 2) Mixing is a chaotic process (the reviewer even suggests potentially turbulent!). Turbulence would destroy substantially any compositional diversity that may emerge in a cooling magma. Lengthscales of homogenization become increasingly small with increasing Reynolds number. Thus, turbulence is not the solution irrespective of this problem. Chaotic fluid flow -a more common assumption for most magmas, especially in a narrowly-confined conduit, would potentially preserve compositional diversity in the melt that could be recorded by growing crystals. However, given that mixing is a chaotic process it also intrinsically produces a multitude of crystal pathways through the mixing regime (e.g. Ruprecht et al. 2008, Schleicher and Bergantz 2017). These pathways vary in space and time along a continuously changing melt composition that becomes variably hybridized. Thus, crystals will record a non-uniform melt composition evolution with some crystals spending significant time in an increasingly primitive magma (i.e. long episodes of reversed zoning), while others spend more time in a magma that is becoming increasingly evolved (i.e. long episodes of normal zoning). Everything in

between is possible, and the result should be a complex crystal cargo where end-member cases and all intermediate stages exist.

We note that our crystal histories while complex have a uniform directionality to them and therefore all crystals experience a similar history of melt evolution (likely modulated by boundary layer effects as stated in our manuscript!).

[Redacted]

Lastly, if we borrow from the recent work of Gordeychik et al. 2018 (their supplementary figure SM2.1b on the left), they demonstrate for some Shiveluch crystal what they consider a mixing trajectory in Ni versus Fo. The compositional path in olivine that is consistent with magma mixing has both an increase in Fo and Ni. This has been observed in Shiveluch but also other volcanoes

such as Llaima and Parícutín. In contrast we see in the Los Hornitos olivines a decrease in Fo associated with the Ni reversals (as shown in Fi. 3 b, c in the manuscript).

Compared to Shiveluch, the Los Hornitos olivines follow a very restricted evolution line along the peridotite source that is consistent with general olivine crystallization from a single evolving magma batch.

In summary, mixing (and definitely mixing in a conduit) is NOT the solution for the kinds of crystals we observe here!

In fact, the authors also explain the oscillatory zonation of P to be due to kinetic effects in a boundary layer. This interpretation must imply rapidly changing growth rates, which then implies rapidly changing temperature conditions. This is unless constitutional undercooling is implied here, which is unlikely because small contents of Al and P do not affect the olivine solidus temperature of the melt. If the kinetic effect is due to growth rate (and thus bulk temperature) changes, then the thermal history of the melt must be highly variable at short time intervals. Conductive processes within the bulk of the melt should then be ruled out and this only leaves (in my opinion) mixing between differently cooled melt batches that form during rapid (even turbulent ??) flow during ascent through a narrow conduit. This may be accompanied by flow differentiation to cause minor changes in melt compositions of mixing magma batches.

R: We are confused whether there is some misunderstanding on this type of boundary layer. It does NOT form and disappear because of large temperature changes, but because diffusive transport to the crystallizing interface becomes modulated by episodes of rapid growth and the changing delivery of melt components to the crystal. Once growth slows down in response to that, the boundary melt layer can relax and normal growth resumes until another episode commences when the boundary layer effects start to overwhelm the system.

These melt portions also would all be slightly different, which would explain the different Fo-contents in the different “frames” (authors second argument).

R: We think the compositional variability is uni-directional and small, consistent with an evolving magma, but not one that contains abundant

different compositional pockets locally at the same time. See our explanation above.

The third argument by the authors is that all crystal types occur in all parts of the tephra deposit even though they nucleated sequentially. In my opinion, this is not at all in conflict with the fast mixing in a conduit. In such scenario olivine crystals would indeed be forming all the time from the different melt batches so crystals from all times and stages of nucleation, skeletal growth and maturation would form and be erupted together.

R: Yes, but actually you would expect also some crystals to form with low Fo cores that then become increasingly high Fo towards the rim, because as the reviewer stated crystals would form “all the time from different melt batches”. We do not see any crystals with reverse zoning in their core.

I still can agree with the authors that in their case, magma mixing may not necessarily be the fundamental process that results in concentric zonation. So their conclusions that “The presence of multiple magma batches stored in the crust to account for several compositional reversals recorded in olivine phenocrysts are not necessarily required” may still be justified. But I would like to see additional arguments !

If their case can be a bit better argued, then this example will firmly establish an alternative interpretation, break the standard assumptions about concentric growth caused by magma mixing and should be always accepted as an alternative explanation for concentric zonation.

A minor fine-tuning of the text here and there to address these issues should be easily done and not require too much effort, but it would be worth doing it.

Wider implications

In my review I suggested to explore the wider implications of this study. This concerns the question whether arc magmas, due to their higher water contents, their different ascent and magma plumbing system in different crustal stress regimes may – or may not – systematically be different with respect to the thermal history and olivine crystallization during the passage of basalt magma from source to surface. I am now more than happy with the extended discussion, this has been done extremely well.

Again, none of these comments should be taken to suggest that this work should not be accepted to Nat Comm. Additional improvements will make this an important contribution to our understanding of ascent and crystallisation of basaltic arc magmas and their potential contrast to other settings.

25.1. 2021

Gerhard Wörner

Reviewer #3 – Anonymous reviewer

We appreciate the comments of the reviewer and we agree with the fact that paired cases or either multiple cases have to be made in other primitive monogenetic vents

not only on the Andes but through other arc olivines and test the sequential versus the out-of-sequence growth. We hope to contribute with this article so that the community research new options when evaluating crystal growth dynamics and basalt transport at arc settings.